# Full-length transcript alterations in human bronchial epithelial cells with *U2AF1* S34F mutations

Cameron M Soulette[1], Eva Hrabeta-Robinson[2] ⓘ, Carlos Arevalo[1], Colette Felton[2] ⓘ, Alison D Tang[2], Maximillian G Marin[2], Angela N Brooks[2] ⓘ

*U2AF1* is one of the most recurrently mutated splicing factors in lung adenocarcinoma and has been shown to cause transcriptome-wide pre-mRNA splicing alterations; however, the full-length altered mRNA isoforms associated with the mutation are largely unknown. To better understand the impact *U2AF1* has on full-length isoform fate and function, we conducted high-throughput long-read cDNA sequencing from isogenic human bronchial epithelial cells with and without a *U2AF1 S34F* mutation. We identified 49,366 multi-exon transcript isoforms, more than half of which did not match GENCODE or short-read–assembled isoforms. We found 198 transcript isoforms with significant expression and usage changes relative to WT, only 68% of which were assembled by short reads. Expression of isoforms from immune-related genes is largely down-regulated in mutant cells and without observed splicing changes. Finally, we reveal that isoforms likely targeted by nonsense-mediated decay are down-regulated in *U2AF1 S34F* cells, suggesting that isoform changes may alter the translational output of those affected genes. Altogether, our work provides a resource of full-length isoforms associated with *U2AF1 S34F* in lung cells.

## Introduction

Previous cancer genomic studies across lung adenocarcinoma (ADC) patients have revealed recurrent mutations in the splicing factor *U2AF1* (Brooks et al, 2014; Cancer Genome Atlas Research Network, 2014; Campbell et al, 2016). U2AF1 is an essential splicing factor that functions to identify the 3′ end of intronic sequence in the early steps of pre-mRNA splicing (Shao et al, 2014). In ADC, the most recurrent *U2AF1* mutation occurs at amino acid residue 34, in which a C > T transition causes a change from serine to phenylalanine (S34F). The impact of *U2AF1 S34F* on pre-mRNA splicing has been widely studied (Przychodzen et al, 2013; Brooks et al, 2014;

Coulon et al, 2014; Ilagan et al, 2015; Shirai et al, 2015; Park et al, 2016; Yip et al, 2017; Palangat et al, 2019; Smith et al, 2019), and previous work has shown that mutant U2AF1 has an altered binding affinity with its pre-mRNA substrate (Okeyo-Owuor et al, 2015; Fei et al, 2016). In ADC, altered binding affinity of mutant U2AF1 has been shown to alter pre-mRNA splicing and other post-transcriptional processes (Brooks et al, 2014; Fei et al, 2016; Park et al, 2016; Palangat et al, 2019).

The impact of *U2AF1* mutations on the transcriptome raises interesting hypotheses for an oncogenic role through mRNA dysregulation. *U2AF1 S34F* is known to alter alternative splicing and polyadenylation of cancer-relevant genes (Przychodzen et al, 2013; Brooks et al, 2014; Ilagan et al, 2015; Okeyo-Owuor et al, 2015; Shirai et al, 2015; Yip et al, 2015, 2017; Fei et al, 2016; Park et al, 2016; Smith et al, 2019). For example, *U2AF1 S34F* perturbs pre-mRNA splicing of interleukin-1 receptor–associated kinase 4 (*IRAK4*) toward producing isoforms that promote activation of kappa-light-chain-enhancer of B cells (*NF-kB*), a factor known to promote leukemic cell growth (Smith et al, 2019). In addition to splicing-dependent functions of *U2AF1 S34F*, recent studies show other effects of *U2AF1 S34F* on modulating translation (Palangat et al, 2019; Akef et al, 2020), formation of R-loops (Chen et al, 2018; Nguyen et al, 2018; Cheruiyot et al, 2021), and affecting the nonsense-mediated decay (NMD) pathway (Cheruiyot et al, 2021). Although some oncogenic roles for *U2AF1 S34F* have been described, the full functional impact of *U2AF1*-associated mRNAs in a lung tissue context is unknown.

Investigating mRNA isoform function proves difficult given the complexity and accuracy of isoform assembly with short reads (Engström et al, 2013; Steijger et al, 2013). Accurate isoform assembly is important in investigating RNA processing alterations associated with global splicing factors, like U2AF1. Recent studies have shown the utility of long-read approaches in capturing full-length mRNA isoforms, by constructing isoforms missed by short-read assembly methods (Oikonomopoulos et al, 2016; Byrne et al, 2017; de Jong et al, 2017; Workman et al, 2019; Tang et al, 2020). Moreover, long-read approaches have already been conducted using RNA derived from primary tumor

---

[1]Department of Molecular, Cellular and Developmental Biology, University of California, Santa Cruz, CA, USA   [2]Department of Biomolecular Engineering, University of California, Santa Cruz, CA, USA

Correspondence: anbrooks@ucsc.edu
Maximillian G Marin's present address is Department of Systems Biology and Department of Biomedical Informatics, Harvard Medical School, Boston, MA, USA

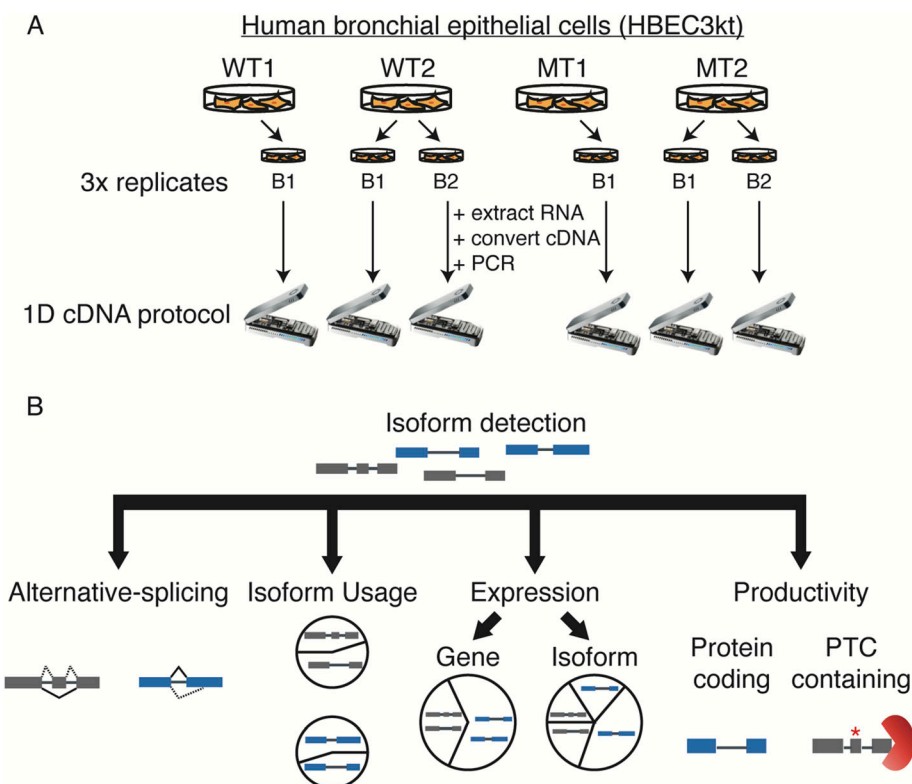

**Figure 1. Full-length isoform sequencing and analysis workflow.**
**(A)** Diagram of experimental setup and sequencing strategy. RNA was extracted from whole cell lysate and converted to cDNA using a poly(A) tail selection strategy. WT and mutant conditions were sequenced in triplicate. Each sequencing run was conducted in parallel, in which a WT or mutant was sequenced on separate row cells. **(B)** Data processing pipeline workflow. Full-Length Alternative Isoform analysis of RNA was used to construct a reference transcriptome from long-read data with matched short-read RNA-seq and to perform differential expression and productivity analyses.

samples harboring *SF3B1* mutations, demonstrating its effectiveness in capturing mutant splicing factor transcriptome alterations (Tang et al, 2020). In addition, studies have shown the extent to which long-read data can be used as a quantitative measure for gene expression (Oikonomopoulos et al, 2016; Byrne et al, 2017). Given the global impact of *U2AF1* mutations on the transcriptome, identifying RNA processing alterations at the level of full-length mRNA isoforms is an essential step in understanding the functional impact of affected mRNAs.

Here, we used a long-read sequencing approach to characterize isoform structure and predicted function of transcript isoforms affected by *U2AF1 S34F*. We have chosen to study *U2AF1 S34F*–associated isoform changes in an isogenic cell line, HBEC3kt cells, which has previously been used as a model for identifying transcriptome changes associated with *U2AF1 S34F* (Ramirez et al, 2004; Fei et al, 2016). We constructed a long-read transcriptome that contains substantial novel mRNA isoforms not reflected in annotations or could they be reconstructed using short-read sequencing assembly approaches. Our long-read data support a strong *U2AF1 S34F* splicing phenotype, in which we demonstrate the ability to recapitulate the splicing phenotype associated with *U2AF1 S34F* mutants using splicing event-level analyses. We found an overall trend for isoform down-regulation, in which isoforms containing premature termination codons (PTCs) and immune-related genes were significantly impacted. Finally, we leverage previously published short-read polysome profiling data to associate changes in translation control for genes affected by *U2AF1 S34F*. Our work here provides the first estimate of the extent to which *U2AF1 S34F* splicing alterations impact mRNA function.

# Results

### Long-read sequencing reveals the complexity of the HBEC3kt transcriptome

We first characterized the transcriptome complexity of HBEC3kt cells with and without *U2AF1 S34F* mutation using the Oxford Nanopore MinION platform. We conducted cDNA sequencing on two clonal cell lines, two WT, and two *U2AF1 S34F* mutation isolates (WT1, WT2, MT1, MT2). We obtained three biological replicates for each WT and MT condition by extracting whole-cell RNA from each cell isolate, one growth replicate of WT1 and MT1, and two independent growth replicates from different time points for WT2 and MT2. We converted RNA into cDNA using methods described in previous nanopore-sequencing studies ([Picelli et al, 2013; Byrne et al, 2017]; see the Materials and Methods section) and performed nanopore 1D cDNA sequencing on individual flow cells (Fig 1A). Our sequencing yielded 8.8 million long reads across all six sequencing runs (Table S1), with each run having an average of 1.5 million reads with an average length of 968 bp. We then processed our long-read nanopore data through a Full-Length Alternative Isoform analysis of RNA (FLAIR) (Tang et al, 2020) to construct a reference transcriptome and perform various differential analyses (Fig 1B, see the Materials and Methods section). We constructed a total of 63,289 total isoforms, 49,366 of which were multi-exon and 45,749 contained unique junction sets (Supplemental Data 1 and Supplemental Data 2). Median transcript length and number of exons were like those in GENCODE annotations, and transcripts detected were consistent across replicates (Fig S1A–D and Table S1).

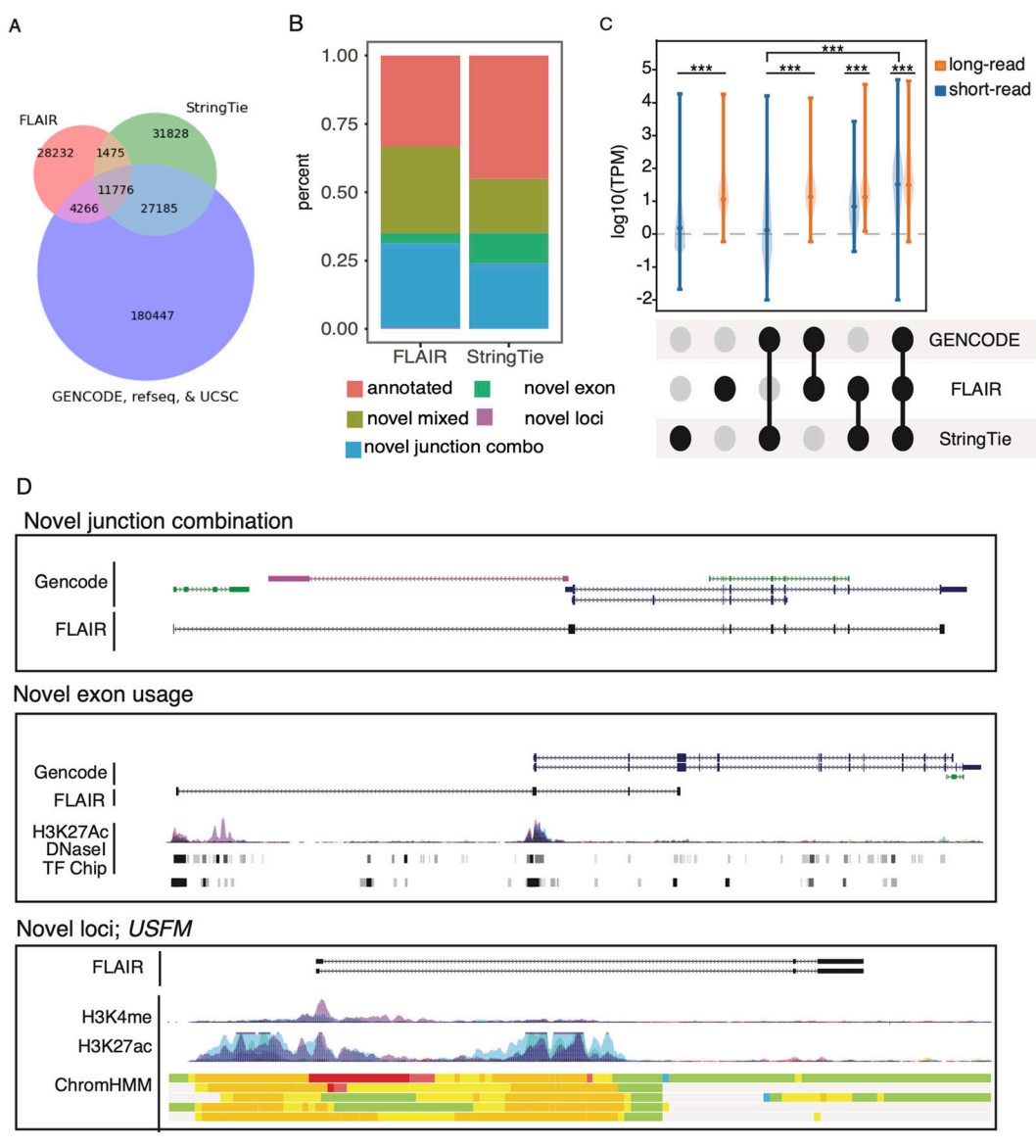

**Figure 2. Full-Length Alternative Isoform analysis of RNA (FLAIR) captures HBEC3kt transcriptome complexity.**
**(A)** Transcript isoform overlap between short-read StringTie assembly, GENCODE annotation, and FLAIR isoforms. **(B)** Isoform annotation categories for FLAIR and StringTie isoforms in comparison to GENCODE v19 annotations. **(C)** Normalized transcript isoform expression levels across overlapping categories between long-read FLAIR, short-read StringTie assembly, and GENCODE annotation. Expression distributions were compared using Wilcoxon rank-sum test, and comparisons denoted with *** have *P*-values <0.001. **(D)** UCSC Genome Browser shot example for novel classification categories. For each panel, GENCODE annotations represent GENCODE v19 basic annotation set. Encode regulatory tracks were included to show H3K27 acetylation, DNase hypersensitivity, transcription factor binding ChIP (TF Chip), and ChromHMM data from various cell lines. Red and yellow hues represent putative promoter regions; green regions represent putative transcribed regions.

We compared long-read FLAIR isoforms against GENCODE reference annotations and a short-read StringTie assembly using previously published data from HBEC3kt cells (Fei et al, 2016); (Supplemental Data 3; see the Materials and Methods section). For comparison to the long-read data, the short-read data were ~100 million paired-end 101 bp reads per sample. We found approximately one–third of our FLAIR transcriptome overlapped with commonly used transcript annotations (GENCODE v19, v33, RefSeq, and UCSC Genes) (Fig 2A). The remaining FLAIR isoforms contained novel elements, such as novel exons, novel junction combinations, or a novel genomic locus (Fig 2B). In contrast, nearly half of the isoforms

from short-read assembly overlapped known GENCODE isoforms. We hypothesized that the increased number of annotated isoforms from short-read assembly could be due to higher sequencing depths. We, therefore, overlapped intron junction chains between all three datasets and quantified expression from each overlapping group. We found a significant difference in the average expression of isoforms exclusive to StringTie relative to FLAIR-detected isoforms (*P*-value <0.01; Fig 2C).

Although our long-read approach did not capture lowly expressed isoforms, we found that FLAIR-exclusive isoforms contained novel exons, junction combinations, and novel loci isoforms

(Fig 2D). Notably, we identified 182 FLAIR-exclusive isoforms from 123 unannotated loci, none of which were assembled by short-reads despite having short-read coverage support, perhaps due to repeat elements that are known to be difficult to assemble across (Treangen & Salzberg, 2011). We investigated a putative long intergenic non-coding RNA (lincRNA) we call *USFM* (up-regulated in splicing factor mutant; *LINC02879*; chr18:26,735,945-26,754,735 [hg19]), which was one of the most highly expressed multi-exon isoforms in mutant samples with 202 reads per million (RPM) (17 RPM in WT; Fig 2D bottom panel and Fig S1E). We manually examined long-reads aligned to *USFM* and found poly(A) tails, suggesting *USFM* supporting reads are not likely to be 3′ end fragmented products. Next, we used publicly available ENCODE data to look for chromatin features that provide additional support for this novel locus (ENCODE Project Consortium, 2012). Peaks associated with H3K27 acetylation and H3K4 methylation suggest the presence of regulated transcribed genomic regions. Moreover, the transcript start site of *USFM* overlapped with active promoter predictions from chromHMM (ENCODE Project Consortium, 2012), an algorithm used to predict promoters and transcriptionally active regions. No significant homology matches to protein-coding (PRO) domains could be found using NCBI BLAST (data not shown). Taken together, these data indicate that *USFM* isoforms have characteristics that are consistent with lncRNAs and highlight the utility of long-reads in identifying putative novel genes.

### *U2AF1 S34F* splicing signature captured by long-read event-level analyses

We compared *U2AF1 S34F*–associated splicing signatures in our long-read data with those found from analyses of short-read datasets (Brooks et al, 2014; Ilagan et al, 2015; Okeyo-Owuor et al, 2015; Fei et al, 2016). Previous reports have shown that cassette exon skipping is the most prevalent splicing alteration induced by *U2AF1 S34F*. Moreover, motif analysis of 3′ splice sites adjacent to altered cassette exons with enhanced and reduced inclusion show a strong nucleotide context for "CAG" and "TAG" acceptor sites, respectively (Brooks et al, 2014; Ilagan et al, 2015; Okeyo-Owuor et al, 2015; Fei et al, 2016). We processed all available RNA-Seq data from The Cancer Genome Atlas (TCGA) from lung ADC tumor samples with *U2AF1 S34F* mutations (n = 11) and without mutations in commonly mutated splicing factors (n = 451) (Table S2). In addition, we analyzed previously published short-read RNA-seq data from HBEC3kt isogenic cell lines (Fei et al, 2016) (Table S3). We observe minor differences in alternative donor and intron retention events between TCGA and HBEC3kt short read, which could likely be explained by the limitations of statistical testing with only two replicates for the HBEC3kt short-read data (Hansen et al, 2011). More notably, our results are consistent with previous reports showing that cassette exon events are the most predominant patterns of altered alternative splicing in TCGA and HBEC3kt data (51/59 and 192/257, respectively).

We next asked if our long-read recapitulates known changes in splicing associated with mutant U2AF1. To do this, we used FLAIR-diffSplice, a FLAIR module that identifies and quantifies alternative splicing events from isoform annotations. We then used DRIMseq (Nowicka & Robinson, 2016) to identify significant changes in percent spliced in values (PSI) and compared the proportions of each AS event type between long-read and short-read sequencing datasets (Fig 3A and Tables S4 and S5; see the Materials and Methods section). We identified 115 significantly altered splicing events, 71 of which had a substantial change in PSI (PSI >10%; Table S6). The most predominant altered splicing type was cassette exon usage (55/71), in which nearly all events showed a decrease in PSI (Fig S2A). The predominance of altered cassette exon usage was also observed when identifying full-length transcripts from a different analysis tool, StringTie2 (Kovaka et al, 2019) (Fig S1F), suggesting our results are robust to different computational methods used. In addition, we observed strong correlation between short- and long-read PSI values from cassette exon events with significant PSI changes (union between datasets; 68 events), suggesting that splicing quantification between technologies is consistent (Fig 3B; Spearman $\rho$ = 0.8). Last, we investigated the 3′ splice site motif associated with altered cassette exons and alternative acceptor events and found "TAG" and "CAG" motifs associated with acceptor sites with reduced and enhanced inclusion, respectively (Fig 3C). Overall, our results demonstrate consistent splicing signatures associated with mutant U2AF1 between short- and long-read methodologies.

*U2AF1 S34F* has been implicated in widespread altered poly(A) site selection (Park et al, 2016). We took advantage of the long-read data to identify poly(A) cleavage sites and identified alternative poly(A) alterations associated with *U2AF1 S34F* (Fig 3D). Identifying poly(A) sites with short-reads is computationally difficult because of alignment of reads primarily composed of poly(A) sequence or alignment across repetitive sequence commonly found in 3′ untranslated regions (Chen et al, 2009; Elkon et al, 2013; Shenker et al, 2015; Ha et al, 2018). We first investigated the presence of poly(A) cleavage site motifs at the 3′ ends of FLAIR isoforms and found a strong signal ~20 nucleotides upstream from transcript end sites for the most commonly used cleavage motif, "AATAAA," relative to random six-mer (Fig S2B). We next tested for alternative poly-adenylation (APA) site usage alterations by comparing the proportion of poly(A) site usage for each gene between *U2AF1* WT and S34F (see the Materials and Methods section). 10 genes demonstrated significant changes in polyadenylation site usage (corrected *P*-value <0.05 and ΔAPA >10%), which comprises 7.2% of all RNA processing alterations identified in this study (11 APA and 142 alternative splicing events), far less than previous reports. The most significant APA alteration occurred in *BUB3* (Fig 3E), which is part of the mitotic checkpoint pathway, a pathway containing genes that are commonly altered in selected lung cancers (Takahashi et al, 1999; Haruki et al, 2001). Collectively, our event-level analyses confirmed our ability to capture well-documented *U2AF1 S34F*–associated splicing signatures with long-read data.

### Long reads provide isoform context for *UPP1*-splicing alterations missed by short-read assembly

We compared the exon connectivity of cassette exons altered by *U2AF1 S34F* in uridine phosphorylase 1 (*UPP1*), which was the most significantly altered gene in our event-level analysis. *UPP1*-altered cassette exons accounted for 4 of the 55 significantly altered cassette exons (exons 5, 6-long, 6-short, and 7), one of which, exon 7,

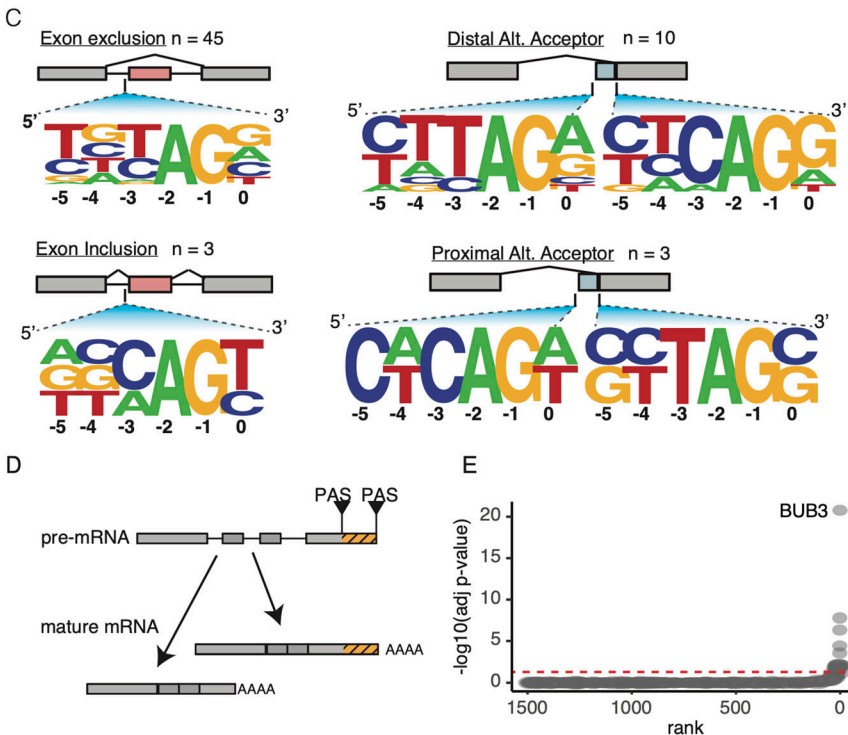

**Figure 3. Nanopore data recapitulates U2AF1 S34F splicing signature.**
**(A)** Alternative splicing events that were found to be significantly altered between WT and U2AF1 S34F conditions. Events are broken down into different patterns of alternative splicing. **(B)** Change in percent spliced in correlation between short- and long-read cassette exon events. **(C)** Motifs of 3′ splice sites for altered cassette exons (left panels) and alternative acceptor sites (right panels) identified using nanopore data. **(D)** Alternative polyadenylation site selection schematic. **(E)** Ranked genes with significant changes in alternative polyadenylation site usage.

was also found to be significantly altered in TCGA ADC data. We compared 28 FLAIR isoforms containing exon 7 against StringTie assembly to determine which isoforms were missed by either method. Despite minor differences in transcript start and end sites, we found all four short-read assembled *UPP1* isoforms containing exon 7 in our set of FLAIR isoforms (Fig S2C). The additional 21 FLAIR-exclusive isoforms contained a mixture of exon skipping events, alternative 3′, and alternative 5′ splicing events that coincided with exon 7 inclusion (Fig S2D top panel). A broader comparison of all 95 *UPP1* FLAIR isoforms revealed that only seven were assembled by short read. We then asked if any of the FLAIR-exclusive *UPP1* isoforms were expressed at substantial proportions (>5%) by

quantifying the expression of each isoform using our long reads. We found that six of the seven most highly expressed *UPP1* isoforms were FLAIR-exclusive (Fig S2D bottom panel). Taken together, although short-read methods assembled complex splicing regulation observed in *UPP1*, our long-read analysis revealed extensive isoform diversity not captured by short reads.

## U2AF1 *S34F* induces strong isoform switching in *UPP1* and *BUB3*

We next assessed transcriptome-wide changes in isoform usage using our long-read data. Short-read event-level analyses typically represent isoforms by distinct RNA processing events such as

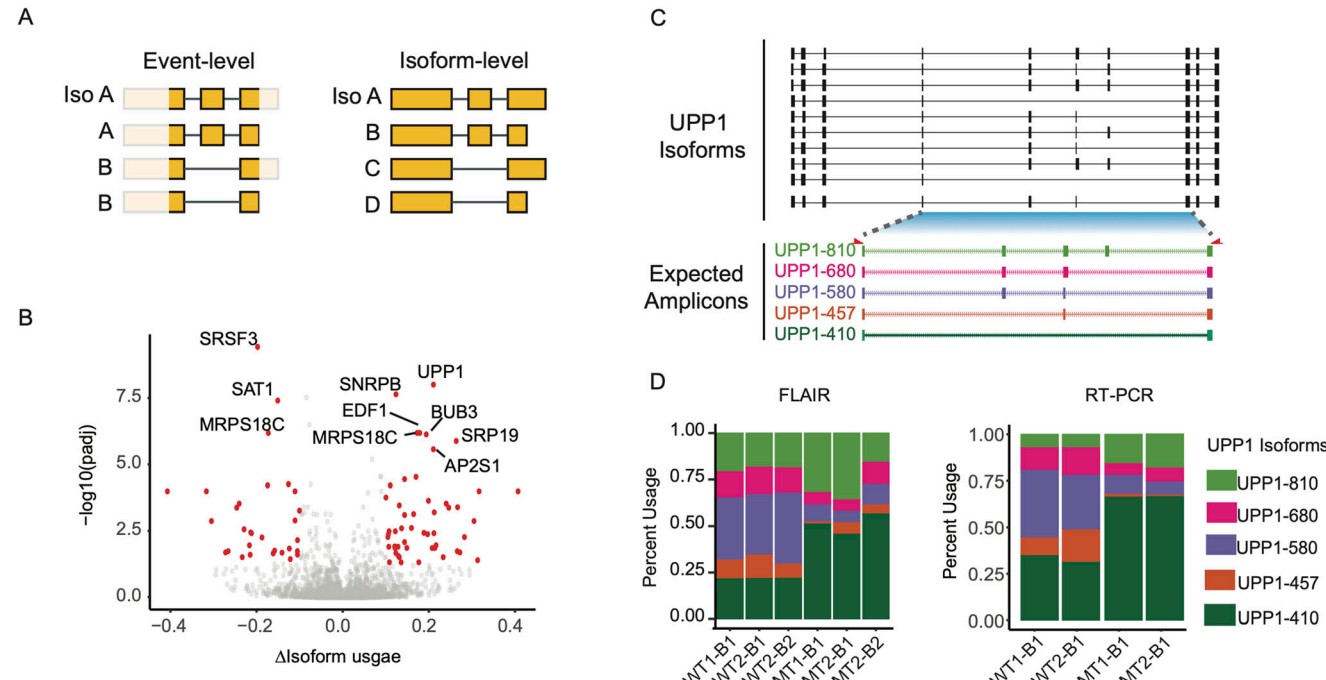

**Figure 4. U2AF1 S34F–associated full-length isoform usage changes.**
**(A)** Diagram of event-level versus isoform-level analyses captured by long-read sequencing alterations. **(B)** Volcano plot of differentially used isoforms. Red dots indicate usage changes with corrected *P*-value <0.05 and magnitude change >10%. Gene names indicate top 10 genes with significant isoform changes. **(C)** UPP1 Full-Length Alternative Isoform analysis of RNA major isoforms (top panel) and predicted amplicons (bottom panel). Isoform numbers correspond to predicted amplicon sizes. Red arrows below major isoforms represent PCR primers used for RT–PCR validation. **(D)** Long-read isoform usage quantified by nanopore data (left panel) and RT–PCR (right panel).

splicing or poly(A) site usage. Long reads capture entire mRNA isoforms and therefore can be used to accurately quantify distinct isoforms (Fig 4A). We tested for changes in both (i) the ratio of isoform expression within a gene (isoform usage) and (ii) the absolute level of expression (isoform expression). We identified 166 isoforms with significant usage changes (corrected *P*-value <0.05) using DRIMSeq, nearly half of which had substantial changes in frequency (82/166; Δisoform usage >10%) (Fig 4B) and Table S7, see the Materials and Methods section). Consistent with our event-level analysis, our isoform usage and expression analysis identified both *BUB3* and *UPP1* in the top 10 most significantly altered genes, suggesting that changes in these isoforms are likely due to splicing changes. We conducted follow-up RT–PCR validation for seven of the top hits from our analysis, six of which showed consistent changes in isoform usage (Fig S3).

We found complex 3′ end processing patterns that define *BUB3* isoforms. Previous reports have described alternative acceptor site usage for *BUB3* that leads to the usage of distinct polyadenylation sites (Bava et al, 2013). Consistent with previous reports, we find that the proximal acceptor site leads to the production of isoforms using three APA sites (APA1, 2, and 3), and usage of a distal acceptor site leading to the usage of two APA sites (APA 4 and 5) (Fig S4A). Our event-level analyses revealed a significant shift toward the usage of the distal acceptor site and APA site (Δalt. acceptor >30%; corrected *P*-value <0.05), which is consistent with differences in isoform usage (Fig S4B). Notably, the proximal *BUB3* 3′ acceptor site is preceded by a thymidine residue, which could partially explain isoform shifting

toward the usage of the distal acceptor site. We also observed a preference for APA site 2 in WT samples (ΔAPA usage 20%), which seems to be lost in mutant samples (ΔAPA usage 6%; two-sided *t* test *P*-value <0.05). Long-read sequencing allowed us to detect and quantify the coupling of alternative 3′ splice site usage with specific APA sites.

We next investigated significant isoform usage changes in *UPP1* (Fig 4C). Out of the 95 *UPP1* isoforms identified by our data, 68 (71%) fell below 1% of the total *UPP1* gene abundance, indicating that most of the isoforms are minor isoforms (Fig S2D). The remaining 28 *UPP1* isoforms were tested for differential isoform usage, two of which were found to have significant usage changes (corrected *P*-value <0.05 and Δisoform usage >10%). RT–PCR validation using primers that span all *U2AF1 S34F*–associated cassette exons showed a pattern consistent with sequencing results, in which *U2AF1 S34F* induces a shift toward *UPP1* isoforms that either contain or exclude all cassette exons (Figs 4D and S2C). *UPP1* is known to be highly expressed in solid tumors (Liu et al, 1998; Kanzaki et al, 2002), but cancer-associated splicing alterations have not been described.

### Isoforms changes are partially explained by event-level splicing changes

We next determined the extent to which *U2AF1 S34F* alters the expression of individual isoforms. This analysis complements our isoform switching analysis by allowing for the identification

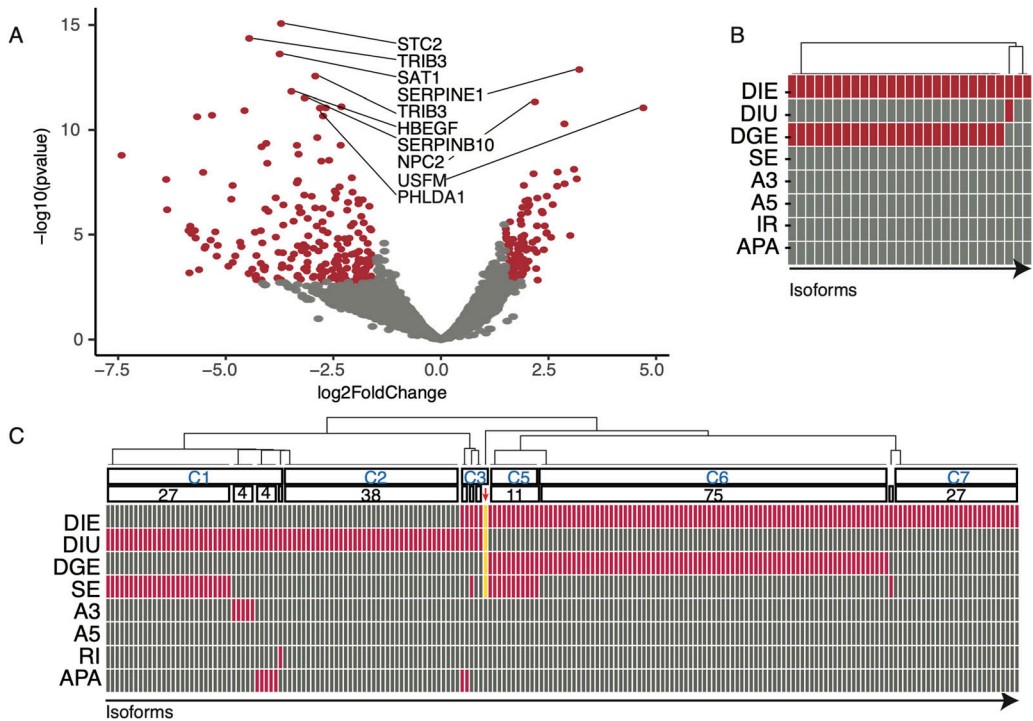

**Figure 5. S34F-associated full-length isoform expression alterations.**
**(A)** Volcano plot of differentially expressed isoforms. Red dots indicate expression changes with adjusted *P*-value <0.05 and magnitude change >1.5. Gene names are ordered by top 10 most significantly altered isoforms. **(B)** Differential event overlap of isoforms from genes involved kappa-light-chain-enhancer of B cells signaling pathway. Each box indicates an isoform, where red signifies if a particular isoform resulted as significantly altered for the corresponding analysis. DIE, differential isoform expression; DIU, differential isoform usage; DGE, differential gene expression; SE, skipped exon; A3, alternative 3′ splice site usage; A5, alternative 5′ splice site usage; IR, intron retention; APA, alternative polyadenylation site usage. **(C)** Same as panel (B), except including all isoforms with altered expression or usage. Red arrow highlights UPP1 (gold bars).

of minor isoforms (isoform usage <10%) with large expression changes, genes with uniform isoform expression changes, or one predominant isoform driving total gene expression changes. Our analysis yielded 122 isoforms with significant changes in expression (corrected *P*-value <0.05 and $log_2$FoldChange >1.5; Fig 5A and Table S8). We found the most up-regulated isoforms were from the putative lincRNA *USFM* ($log_2$FoldChange >3 and corrected *P*-value <0.01). We searched TCGA ADC short-read RNA-seq data for expression of *USFM*, but we could not find substantial read counts (<1 RPM) for samples with or without *U2AF1 S34F* mutations. LincRNAs tend to be expressed at lower levels than PRO genes (Djebali et al, 2012), which may explain why *USFM* was not detected at the depth of sequencing of TCGA samples.

We conducted a gene set enrichment analysis on differentially expressed isoforms and found that isoforms from NF-kB via TNF signaling pathway were significantly down-regulated (corrected *P*-value <0.05; Table S8). This observation is consistent with recent reports, in which *U2AF1 S34F* has been shown to modulate immune-related pathways (Palangat et al, 2019; Smith et al, 2019). We then asked if *U2AF1 S34F*–associated splicing alterations could explain expression changes in isoforms from genes involved in NF-kB signaling pathway by overlapping our event-level splicing analyses along with our gene and isoform expression analyses (Fig 5B). We observed no splicing alterations that could explain significant changes in expression or isoform usage, suggesting that the

expression of these isoforms may be modulated through a splicing-independent mechanism or by splicing alterations not detected by sequencing. Most of these isoform changes were associated with total gene expression changes (Fig 5B), suggesting these are transcriptionally regulated.

We expanded our alternative splicing overlap analysis to ask which of the 198 isoforms with altered usage or expression coincided with other significantly altered features, such as alternative splicing and gene expression. We found several clusters of features that partially explain the involvement of *U2AF1 S34F* mutation in isoform expression and usage dysregulation. For example, we found 27 isoforms with both significant isoform usage and cassette exon usage changes (Fig 5C cluster C1). It is possible that these 27 isoforms with significant usage changes are defined by single exon skipping events and are likely directly induced by *U2AF1 S34F*. In contrast, we found several isoforms that did not overlap any other altered features (Fig 5C clusters C2 and C7). We suspect these isoform changes to be modulated through either a splicing-independent mechanism or splicing changes undetected by long reads. Similar to isoforms in the NF-kB signaling pathway, we found a total of 75 isoforms with differential isoform expression and total gene expression (Fig 5C cluster C6). We found a single gene, *UPP1*, that contained four overlapping features, which were changes in isoform expression, gene expression, cassette exon usage, and isoform usage. Altogether, we observed a consistent pattern of

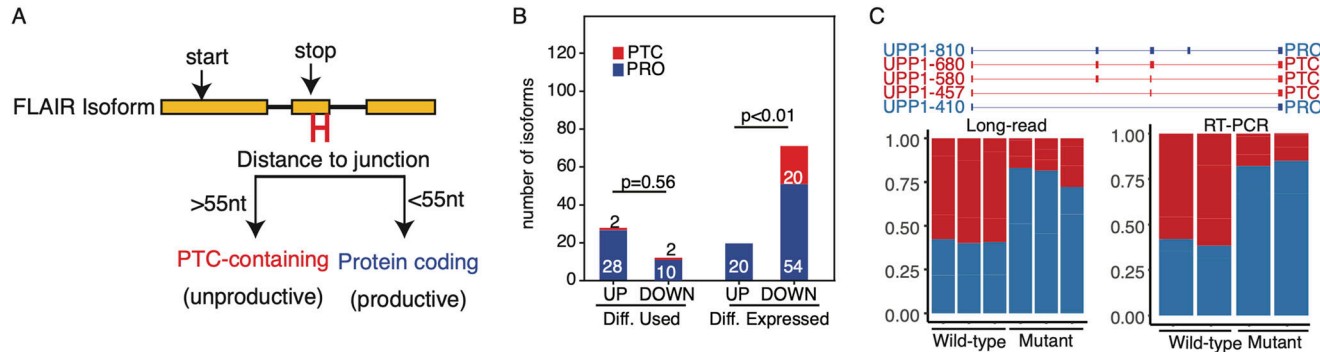

**Figure 6.   U2AF1 S34F induces shifts in isoform productivity.**
**(A)** Diagram of isoform productivity logic prediction. **(B)** Comparison of up- and down-regulated S34F-associated isoform changes classified by productivity. UP, up-regulated; DOWN, down-regulated. Up-regulated indicates isoforms with increased usage frequency or expression relative to WT. Down-regulated indicates isoforms with decreased usage frequency or expression relative to WT. **(C)** UPP1 major isoforms classified by productivity. Bar plots show quantification of each productivity type for nanopore long-read data (left panel) and RT–PCR quantification (right panel).

*UPP1* alterations associated with *U2AF1 S34F*, and also identified populations of dysregulated isoforms that may be modulated through splicing-dependent and -independent mechanisms.

### PTC-containing isoforms are down-regulated by *U2AF1 S34F*

Our long-read approach enables a more confident ORF prediction, which can be used to identify altered splicing events that trigger NMD (Tang et al, 2020). NMD is a process that removes erroneously spliced mRNAs with truncated ORFs that could give rise to gain-of-function or dominant-negative protein products (Dreyfuss et al, 2002; Lewis et al, 2003; Sterne-Weiler et al, 2013; Maslon et al, 2014; Floor & Doudna, 2016; Aviner et al, 2017), and splicing alterations associated with cancer-specific splicing factor mutations have been shown to be substrates of NMD (Yip et al, 2017; Rahman et al, 2020). We therefore asked what fraction of S34F-associated isoform alterations could be putative NMD targets. To do this, we classified FLAIR isoforms into two categories, either as putative PRO isoforms or PTC-containing isoforms (see the Materials and Methods section; Fig 6A). We postulated that the shallow sequencing depth of long-reads relative to short-reads would limit our ability in capturing PTC-containing isoforms if they are indeed subject to NMD. However, of our 63,289 FLAIR isoforms, we identified 8,037 PTC-containing isoforms (12% of all isoforms). We then asked what proportion of PTC-containing isoforms is dysregulated at the level of expression and isoform usage (Fig 6B). For differentially used isoforms, we found similar proportions of PTC-containing isoforms (Fisher's exact two-sided test, *P* = 0.5). In contrast, we found a significant difference in the proportion of PTC-containing isoforms between differentially expressed isoforms (Fisher's exact two-sided test, *P* < 0.01).

We next sought to test if *U2AF1 S34F* induces shifts in isoform productivity by conducting a gene-level analysis. To do this, we compared the proportion of PTC-containing versus productive isoform usage for each gene using the same methodology as our differential isoform usage analysis. Our results showed very few genes with strong shifts in productivity (Fig S5A; total of 10 with corrected *P*-value <0.05 and ΔPTC isoform usage >10%). However, we did

identify a very strong shift in productivity in *UPP1* (*P*-value <0.001 and Δproductivity >20%), a gene we found to have strong changes in splicing, isoform usage, and expression. Interestingly, our differential gene expression analysis showed significant down-regulation for *UPP1* (log₂FoldChange 2.4 and *P*-value <0.05), yet our productivity analysis showed a strong shift toward productive isoform usage (Fig 6C).

### *U2AF1 S34F* isoform dysregulation is associated with changes in translation

We predicted that if PTC-containing *UPP1* isoform are indeed subject to NMD, then the proportion of *UPP1* mRNAs able to undergo translation will be larger in mutant cells relative to WT given the associated shift toward productive isoforms. To test this, we used publicly available polysome profiling data from HBEC3kt cells with and without *U2AF1 S34F* causing mutations (see the Materials and Methods section; Fig S5B). We found a significant change in the proportion of *UPP1* expression across different polysome fractions (chi-squared *P*-value <0.01; Fig S5C). We observed a large drop (Δ10%) in the proportion of expression in polysome fractions five and six between mutant and WT. These fractions correspond to the monosome, which is a fraction not associated with active translation, and is known to harbor non-coding mRNAs, such as NMD products (Floor & Doudna, 2016). The marked shift of *UPP1* expression in mutant samples from the monosome toward higher polysome (fractions ≥ 7) is consistent with the hypothesis that *U2AF1 S34F*–associated *UPP1* alterations alter mRNA fate by shifting isoform production toward isoforms associated with enhanced translational activity.

We next tested if *U2AF1 S34F*–associated isoform changes in *BUB3* are consistent with differences in polysome profiles. In contrast to *UPP1*, we did not observe significant isoform productivity changes for *BUB3*. Instead, we observed significant changes in a terminal alternative 3′ splice site event, that is, linked to APA site usage. Previous reports show that *BUB3* APA site 5 is associated with enhanced translational efficiency (Bava et al, 2013). Our APA analysis showed mutant-specific isoform shifts toward isoforms with APA site 5, effectively increasing the proportion of

translationally efficient *BUB3* isoforms. We tested for changes in *BUB3* polysome profiles using the same methodology used for *UPP1*. We found a strong shift in *BUB3* expression toward high polysome fractions (Fig S5D; chi-squared *P*-value <0.01). Notably, RNA-IP results from previous reports do not support large changes in cytosolic U2AF1 binding for *BUB3* or *UPP1*, which is a proposed mechanism of mutant *U2AF1* to modulate translational efficiency (Palangat et al, 2019).

We next determined if changes in translational control is a general feature for genes with strong changes in isoform expression and usage. To do this, we compared the frequency of genes with significant shifts in polysome profiles between U2AF1 S34F–dysregulated genes identified in this study and genes not found to be dysregulated by mutant U2AF1. We found that 66% (42/63) of genes with mutant-associated isoform changes also had a significant change in polysome profile (see the Materials and Methods section). This proportion was significantly higher (Fisher's two-sided test *P*-value <0.01) than the 48% (1,340/2,753) of genes without S34F-associated isoform changes. When looking more specifically at the 10 genes with changes in APA usage, similar to BUB3, six were associated with strong polysome profile shifts toward higher polysome fractions, three showed a shift toward monosome fractions, and one remained the same across WT and MT conditions (Fig S5E). Altogether, our results are consistent with previous work, implicating *U2AF1 S34F* as a modulator of the translational landscape, yet the lack of RNA-IP support for few cases suggest another possible non-canonical method of translational regulation (Palangat et al, 2019).

## Discussion

In this study, we assessed the impact of *U2AF1 S34F*–associated RNA processing alterations on individual mRNAs using an isogenic cell line harboring a *U2AF1 S34F* mutant allele. Although splicing alterations associated with *U2AF1* have been characterized with short-read sequencing, the full-length isoform context in which the altered events occur has not been described. We aimed to fill this gap in knowledge by using a long-read sequencing approach and supplemented our analysis with orthogonal short-read RNA-sequencing datasets from the same isogenic cell lines.

We demonstrate the robustness of long-read approaches by recapitulating splicing signatures associated with *U2AF1 S34F* mutations. Although our long-read transcriptome captures a comparable number of isoforms relative to short-read approaches, we still lack sequencing depth to capture the entire catalog of cassette exons associated with *U2AF1 S34F*, such as known cassette exons in *STRAP* or *ASUN* which were previously described to have *U2AF1 S34F*–associated splicing alterations (Fei et al, 2016). Moreover, although we identified genes with significant changes in polyadenylation site selection, we were unable to recapitulate transcriptome-wide levels observed in previous studies (Park et al, 2016). In line with these shortcomings, a saturation analysis of full-length isoforms construction reveals isoform discovery limitations, possible due to relatively shallow sequencing depth (Fig S6). However, long-read

sequencing approaches offered by PacBio and Oxford nanopore are continually improving sequencing throughput and quality. Recent studies using newer Nanopore flow cell chemistry and higher-throughput platforms have demonstrated data yield orders of magnitude greater than this study (Tang et al, 2020). With greater data yield and improved transcriptome coverage, there is the potential to identify more *U2AF1 S34F*–dysregulated isoforms with greater confidence.

We observe an interesting link between isoform dysregulation and translational control. Previous studies using RNA immuno-precipitation assays have shown that cytosolic mRNA binding of U2AF1 can modulate translational control (Palangat et al, 2019). This splicing-independent mechanism of translational control is complementary to our findings here, in which isoforms arising from RNA processing alterations caused by *U2AF1 S34F* cause changes in translational control of the gene. Interestingly, our data are consistent with two potential mechanisms. In the case of *BUB3*, *U2AF1 S34F* induces isoform switches toward isoforms with regulatory sequences that promote high translational efficiency. Alternatively, for *UPP1*, we observe a substantial shift away from PTC-containing isoforms, which could serve as putative NMD targets. Although further studies are necessary to directly test if these PTC-containing isoforms are regulated by NMD, we hypothesize that PTC-containing isoforms are strongly selected against in the presence of *U2AF1 S34F*. This could be due to differences in NMD efficiency. Previous studies suggest that NMD is inhibited by U2AF1 mutations (Cheruiyot et al, 2021). If there are global changes in NMD efficiency, our results suggest that there is enhanced NMD efficiency in *U2AF1* mutant cells which is inconsistent with previous findings. This discrepancy may be explained by the difference in cell line context and method of introducing the U2AF1 mutation. Future studies that would complement our analysis here would include NMD efficiency analysis by using a fluorescence-based NMD reported in the presence of mutant *U2AF1* in a lung cell line context.

Our analyses contribute several findings implicating *UPP1* as severely dysregulated by *U2AF1 S34F*. So far, no reports have mentioned isoform-specific dysregulation associated with UPP1. *UPP1* encodes a uridine phosphorylase, which helps maintain homeostatic levels of uridine for RNA synthesis and has been observed to be up-regulated in certain cancer types (Liu et al, 1998). In our study using non-cancer derived cells, we find an opposite pattern, in which *UPP1* is significantly down-regulated at the level of overall gene expression. The observed down-regulation of *UPP1* is consistent with our finding of down-regulation of isoforms involved in the TNF via NF-kB signaling pathway, which is a positive regulator of *UPP1* expression (Wan et al, 2006). However, although we observe a strong down-regulation at the level of total gene expression, our isoform usage and productivity analyses reveal a shift toward more productive isoforms. Nevertheless, further studies are required to determine what impacts *UPP1* isoform changes have on cellular function.

Overall, our data captured the context in which *U2AF1 S34F* RNA processing alterations occur at full-length isoform resolution. We build upon previous short-read analyses by providing an extensive list of isoform-specific changes associated with *U2AF1 S34F*, along with the first estimates of isoform function. Our results demonstrate the importance of investigating the transcriptome of mutant

splicing factors using long-read data that provides diverse perspectives on RNA processing and isoform function.

# Materials and Methods

### Preparing RNA for long-read sequencing

HBEC3kt cells with and without *U2AF1 S34F* were cultured as previously described (Ramirez et al, 2004; Fei et al, 2016). Total RNA was extracted from whole cell lysate using Zymo Direct-zol RNA kits. Purified RNA was prepared for long read following previously established protocols (Picelli et al, 2013; Byrne et al, 2017; Tang et al, 2020). Total RNA was reverse transcribed using the SmartSeq2 protocol and amplified using 15 cycles of PCR. 1 µg of PCR amplified cDNA from each sample was subsequently used for Oxford Nanopore 1D library preparation (SQK-LSK108) on flow cell chemistry version 9.4. Basecalling was performed using Albacore version 2.1.0 using options—flowcell FLO-MIN106 and kit SQK-LSK108. Nanopore reads were prepared for genomic alignment by removing adapter sequenced using Porechop version 0.2.3 (Wick, 2017). After adapter removal, reads were aligned to GENCODE hg19 using minimap2 version 2.14-r894-dirty (Li, 2018) using the "-ax" option.

### Processing TCGA LUAD short-read RNA-seq data

Lung ADC short-read data from TCGA (601 samples total) was downloaded from CGhub using gtdownload (Wilks et al, 2014). TCGA donors with multiple RNA-seq bams were filtered by date to only include the most recent RNA-seq bam (495 samples). 495 TCGA bams were subsequently processed through JuncBASE using default parameters with GENCODE hg19 comprehensive annotations and basic annotations as input to "getASEventReadCounts" for options "--txt_db1" and "txt_db2," respectively (Brooks et al, 2011). Differential splicing analyses were performed using Wilcoxon rank-sum between samples containing *U2AF1 S34F* splicing factor mutation (n = 11) or no splicing factor mutation (n = 451), which were defined by molecular profiling details outlined in the study by Campbell et al (2016).

### Obtaining and processing HBEC3kt short-read RNA-seq data

Short-read HBEC3kt data were retrieved from NCBI short-read archive (GSE80136). Reads were aligned to GENCODE hg19 using STAR version 2.5.3a (Dobin et al, 2013) with parameters "--twopassMode Basic." Aligned bams were subsequently individually used for transcriptome assembly using StringTie version 1.3.5 using GENCODE hg19 basic annotations (Pertea et al, 2015). Individual general transfer formatted annotation files generated from StringTie were then merged using default parameters. For the differential splicing analysis of HBEC3kt short-read data, we used JuncBASE with the same methodology as described in the TCGA LUAD short-read data methods section. HBEC3kt short-read data had two biological replicates per condition (WT and mutant); therefore, for statistical testing, we conducted pairwise Fisher's tests, then defined significant

events as ones with a Benjamini–Hochberg corrected *P*-value >0.05 within each condition and a corrected *P*-value <0.05 between samples across conditions. We then post-filtered significant events to remove redundant and overlapping events by running JuncBASE scripts "makeNonRedundantAS.py" and "getSimpleAS.py." To compare long- and short-read ΔPSI values, we computed PSI changes for significant long-read cassette exon events by subtracting DRIMSeq-calculated proportion values for WT and mutant. We then filtered our short-read JuncBASE PSI table for significant long-read events and computed the short-read change in PSI by subtracting the average PSI between WT and mutant.

### Nanopore read correction, FLAIR-correct

Aligned Nanopore sequencing data were concatenated before running FLAIR v1.4 (Tang et al, 2020) using samtools v 1.9 (Li et al, 2009). Bam files were converted to bed using FLAIR-bam2bed12. Converted bed alignments were subsequently corrected using "FLAIR-correct" with GENCODE hg38 basic annotations. Junctions identified by STAR alignment of HBEC3kt short-read data were also used as input into FLAIR-correct. Briefly, STAR junctions were kept if they contained at least three uniquely aligned sequences either in both Mut1a and Mut1b samples or in both WT1 and WT2 samples. The short-read junctions, along with GENCODE annotated junctions, were used to correct misaligned splice sites in the nanopore data to the nearest site within 10 bp. Junctions that did not follow GT-AG splicing motif were also removed.

### FLAIR-collapse and diffExp

Differential analyses were performed by FLAIR-diffExp with default parameters. Genes and isoforms with less than 10 reads from either sample group were excluded from isoform expression and usage analyses. A merge of GENCODE v19 was provided to FLAIR, which matched detected isoforms to the annotation based on the intron chain.

### Long-read alternative splicing analysis, FLAIR-diffSplice

Differential alternative splicing for long-read data was conducted with FLAIR-diffSplice. FLAIR-diffSplice call events for the following alternative splicing types: cassette exon usage, alternative 3′ splice site, alternative 5′ splice site, intron retention, and APA. PSI values for each event were calculated by tallying the number of reads supporting isoforms that include an event, divided by the total number of reads that span the event. Inclusion and exclusion counts were then constructed into a table to process with DRIMseq (Nowicka & Robinson, 2016) for differential splicing analysis. Differential alternative splicing was identified using FLAIR-diffSplice but with transcripts generated from StringTie2 v2.15 (Kovaka et al, 2019) for comparison. StringTie2 was run with default parameters.

### Long-read APA analysis

Poly(A) cleavage sites were defined by clustering FLAIR isoform transcript end sites using bedtools cluster, with a window distance of five (Quinlan & Hall, 2010; Quinlan, 2014). Poly(A) sites were then

quantified by summing the total number of aligned read counts for each isoform that fell within each cluster. Clusters were assigned to genes, and counts for each cluster were then processed by DRIMSeq. Genes with corrected $P$-value <0.05 were considered to have significant changes in poly(A) site usage.

### Gene set enrichment analysis

The Molecular Signatures Database (Liberzon et al, 2011, 2015) was used to perform all gene set enrichment analysis using gene sets: GO gene sets, Hallmarks, and Canonical pathways. Gene names included for isoform expression and isoform usage analyses were from isoforms with corrected $P$-value <0.05, and magnitude changes of $\Delta Log_2FoldChange$ >1.5 and $\Delta$10% isoform usage. Duplicate gene names from genes with multiple significantly altered isoforms were included only once.

### Polysome analysis

Polysome profiling data from HBEC3kt cells with and without *U2AF1 S34F* mutation were obtained (Palangat et al [2019]; Table S4). For each gene, normalized read counts across polysome fractions three through 10–12 were compared between mutant and WT samples using Chi-squared test. Genes with less than 11 normalized read counts in any given fraction were not tested. Multiple testing correction was conducted using the Python module statsmodels.stats.multitest.multipletests with default parameters. Significant changes in polysome profile were considered to have a corrected $P$-value of <0.05. We tested for general polysome profile alterations in *U2AF1 S34F*–associated genes by comparing the ratio of affected genes with and without significant changes in polysome profile versus unaffected genes. Affected genes were considered ones with either a significant isoform expression or usage change.

### Statistics and significance testing

Results from all differential analyses were called significant if their corrected $P$-value fell below $P < 0.05$ and passed a magnitude filter. For differentially expressed isoforms, events over a $log_2FoldChange$ of 1.5 were called significant. For differentially used isoforms and alternative splicing events, events with ≥10% change in usage were called significant.

### RT–PCR validations

We used RT–PCR to validate splicing an mRNA isoform changes observed in our nanopore data using methods similar to the methods used by Anczuków et al (2012). Reverse transcription was performed using MultiScribe superscript kit A&B Biosystems with 1 $\mu$g bulk RNA. Briefly, a mix containing 1 $\mu$g of bulk RNA and 2 $\mu$l of random 10x primers + $H_2O$ were added to a mixture with a total volume of 13.7 $\mu$l and held at 65°C for 10 min. Next, RT–PCR mixture was added to a master mix containing 2 $\mu$l 10x RT buffer, 0.8 $\mu$l of 25x dNTP mix (100 mM), 1 $\mu$l of RT enzyme, and 0.5 $\mu$l of SUPERase In (20 U/$\mu$l). RT–PCR mix was placed in the thermocycler for 10 min at 25°C, 120 min at 37°C, and 5 min at 85°C. PCR validations were then performed using touchdown PCR, using Titanium Taq in which the annealing temperature was dropped by –1°C for the first 10 cycles, followed by 19 cycles of optimal annealing temperature for each primer set. Cycling conditions were as follows: 95°C/3 min, followed by 10 cycles (95°C/30 s, 70°C/45 s [–1°C per cycle], 72°C/1 min), followed by 19 cycles (95°C/30 s, 58°C/45 s, 72°C/1 min).

List of primers used for touchdown PCR can be found in Supplemental Data 4. PCR products were visualized on an Agilent Tapestation 4150 using D1000 ScreenTape and reagents.

### Code availability

All FLAIR-related scripts and modules used in this study can be found at https://github.com/BrooksLabUCSC/FLAIR. FLAIR commands and other codes are available as Jupyter notebooks upon request.

## Data Availability

Long-read nanopore sequencing data from HBEC3kt WT, and *U2AF1 S34F* cells are available in the NCBI GEO database (GSE140734 accession number).

## Supplementary Information

## Acknowledgements

We thank Dennis Fei and Harold Varmus for providing the HBEC3kt cells. We also acknowledge the ENCODE Consortium and the ENCODE production laboratory of Bradley E Bernstein and the Kellis Computational Biology group for production of the histone regulatory data used via the UCSC Genome Browser tracks for this study. This work was supported by the Pew Charitable Trusts and the University of California Tobacco-Related Disease Research Program T29KT040I (AN Brooks). CM Soulette was supported by training grants NIH T32GM008646, R25GM058903, and the Ford Foundation predoctoral fellowship. AD Tang was funded through NIH grant T32HG008345. C Arevalo was supported by NIH R25HG006836. MG Marin was supported by R25GM058903.

### Author Contributions

CM Soulette: conceptualization, resources, data curation, software, formal analysis, supervision, funding acquisition, validation, investigation, visualization, methodology, project administration, and writing—original draft, review, and editing.
E Hrabeta-Robinson: conceptualization, resources, data curation, software, formal analysis, validation, investigation, visualization, methodology, project administration, and writing—original draft, review, and editing.
C Arevalo: formal analysis, validation, investigation, visualization, methodology, and writing—review and editing.
C Felton: data curation, formal analysis, validation, investigation, visualization, and writing—review and editing.

AD Tang: data curation, software, formal analysis, investigation, visualization, and writing—review and editing.

MG Marin: software and formal analysis.

AN Brooks: conceptualization, resources, formal analysis, supervision, funding acquisition, investigation, methodology, project administration, and writing—review and editing.

## Conflict of Interest Statement

AN Brooks is a consultant for Remix Therapeutics, Inc. All other authors declare that they have no competing interests.

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
