## [Reviewer comments · Life Science Alliance]

Life Science Alliance

Full-length transcript alterations in human bronchial epithelial cells with U2AF1 S34F mutations

Cameron Soulette, Eva Hrabeta-Robinson, Carlos Arevalo, Colette Felton, Alison Tang, Maximillian Marin, and Angela Brooks
DOI: <https://doi.org/10.26508/lsa.202000641>

Corresponding author(s): Angela Brooks, University of California, Santa Cruz

Review Timeline:

Submission Date:	2020-01-08
Editorial Decision:	2020-01-23
Revision Received:	2023-04-13
Editorial Decision:	2023-06-12
Revision Received:	2023-06-30
Accepted:	2023-07-03

Transaction Report:

January 23, 2020

Re: Life Science Alliance manuscript #LSA-2020-00641-T

Dr. Angela Brooks
University of California Santa Cruz
Department of Biomolecular Engineering
Santa Cruz, CA 95060

Dear Dr. Brooks,

Thank you for submitting your manuscript entitled "Nanopore sequencing reveals U2AF1 S34F-associated full-length isoforms". The manuscript has been evaluated by expert reviewers, whose reports are appended below.

As you will see, while the reviewers in principle appreciate the aim of your work, reviewer #3 points out that the robustness of the analysis and your conclusions is unclear at this stage, because more controls (such as spike-ins) and further validation would be needed. Addressing these concerns as well as the other issues raised by all reviewers (some of which require a significant re-analysis) is feasible, but also rather demanding as large parts of the study would need to get repeated. We therefore concluded that we have to return your manuscript at this stage with the message that we cannot publish it here. Having said this, should you be willing to embark into a demanding revision that fully addresses all reviewer concerns, we would be happy to take a look at such a revised version in the future.

Thank you for thinking of Life Science Alliance as an appropriate place to publish your work.

Sincerely,

Reviewer #1 (Comments to the Authors (Required)):

GENERAL ASSESSMENT

Soulette et al describe and analyze a Nanopore-based, long-read sequencing approach to identifying isoforms that are differentially expressed or spliced in the absence or presence of the common U2AF1 S34F mutation. This represents the first published assessment of isoform-level consequences of U2AF1 mutations using long-read technologies, and as such, provides a novel and useful contribution to the field. The data are thoroughly described and analyzed and appear to be of high quality (for example, the high correlation ($r = 0.88$) between delta psi values computed with short-read versus long-read data is impressive given the much lower coverage available for long-read data).

MINOR COMMENTS

1. This manuscript's most important contribution is a database of isoforms in WT and U2AF1-mutant cells, which is presented in GTF format in Supplemental File 1. The utility of this database could be significantly improved with a few simple changes, such as separating out the internal IDs (presumably assigned by FLAIR) from Ensembl gene and transcript IDs and adding gene names from HUGO. The same suggestions apply to the Supplementary Tables. Ensembl gene and transcript ID matching, as well as CDS assignment, should also be fully described in the methods.
2. The comparison to Gencode makes sense given the isoform-level information provided by long-read sequencing and FLAIR. However, the analysis of annotated versus unannotated junctions in Fig. 1 would be strengthened by a comparison to a more comprehensive junction database, such as MISO's isoform annotation.
3. The UCSC Genome Browser view in Fig. 4C is too small to be understandable.
4. Because the experiments were conducted in unperturbed (not NMD-inhibited) cells, an alternative possibility is that NMD efficiency differs between the cell types. This seems unlikely, but nevertheless should be mentioned as a possible caveat when interpreting the data for Fig. 6.

Reviewer #2 (Comments to the Authors (Required)):

The manuscript by Soulette et al, entitled "Nanopore sequencing reveals U2AF1 S34F-associated full-length isoforms" reports long-read RNAseq results from U2AF1 mutant HBEC3kt cells. The authors identified isoforms not detected by short-read RNAseq and highlight how alternative splicing of two genes may affect translational activity.

Major:

1. Please add the level of expression of LINC02879 from long-read data for MT and WT samples.
2. Please include the overlap in junctions between long-read and short-read data for Fig. 3A (225 vs. 142 junctions). Could the authors also provide the overlap between short-read TCGA and short-read HBEC3kt.
3. Can the authors provide a clear explanation in the text for the difference between isoform usage (ratio of isoform expression within a gene) versus the isoform expression (absolute level of expression). This will be important to understand Fig. 5 and the UPP1 results.
4. Fig. 5A and 5C. It appears that the majority of the DIE are associated with DGE in Fig 5B and 5C, suggesting that most of the isoform expression changes are due to gene expression changes and may not be as informative as the isoform usage data. Can the authors comment on this.
5. UPP1 had increased productive isoforms in mutant cells, but significant downregulation of expression. This seems counterintuitive, can the authors suggest a reason? Can the authors provide another gene example of a productive isoform but downregulation of expression? Is this an isolated gene, or is it common in U2AF1 mutant cells?
6. Are there other gene examples of APA shifts that lead to expression changes in polysome fractions? Is BUB3 an isolated example, or is this finding common in U2AF1 mutant cells?

Minor:

1. Please include in the main text the summary statistics for the number of reads and read length per sample for the Nanopore data and the short-read data from HBEC3kt cells.
2. On page 6, the authors hypothesized that the increased number of annotated isoforms from short-read assembly could be due to higher sequencing depths. It is unclear if the results in Fig 2B support their hypothesis. Please provide the answer in the text.
3. How reproducible are the long-read results based on a technical replicate - were the results from MT2 B1 and MT2 B2 samples more similar to each other than MT1 B1?
4. Why does LINC02879 need to be renamed USFM by the authors? Why not just call it LINC02879 to avoid confusion.
5. On the top of page 15, "198 isoforms with altered usage and expression", I think should read "usage or expression" not "and".
6. Fig. 4B. The two green colors are very similar in panel D. Is there an alternative color or style?
7. Fig 5C. Please highlight the UPP1 gene in this panel.
8. In the discussion, please add additional information on why UPP1 is important - what is its normal function, etc.
9. Can the authors please provide more information on the FLAIRcorrect method here?
10. Supp file 3. Please add gene symbols.
11. Supp Table 2. Please add the average or median read counts for each group. Alternatively, provide the read counts per sample if there are not too many samples.
12. Supp Table 3. Please add the mutation status of each sample.
13. Supp Table 4. Please add the gene symbol and read counts.
14. Supp Table 5. Please add values for each sample.
15. Supp Table 6. Please add the gene symbols and values for each sample.

Reviewer #3 (Comments to the Authors (Required)):

SUMMARY

In this work, Soulette and colleagues explore the isoform alterations that result from the mutation U2AF1 S34F, which is a recurrent mutation in lung adenocarcinoma. To this end, the authors conduct

high-throughput long-read cDNA sequencing using nanopore technologies, comparing HBEC3kt cells (WT) to those that contain the mutation (MUT). Using this strategy, the authors report 75% of novel isoforms that do not overlap with GENCODE annotations. They then identify genes with significant isoform usage differences, as well as pursue additional analyses to address the consequences of isoform usage, such as changes in the translational profiles. While the rationale of the work is correct and the question is relevant, I find that there are essential controls that are missing in this work, and without them it is impossible to evaluate whether the findings are correct. Firstly, and perhaps this is the most important point of all, the authors do not include RNA spike-ins (e.g. sequins, SIRVs) in their samples that could serve to assess the false discovery rate of isoforms or to assess the robustness of their bioinformatic pipeline in the analyses of their data. While nanopore technologies are capable of detecting full-length mRNA isoforms, there are plenty of artefacts that can affect the detection of novel isoforms, which can be caused by the library preparation (fragmentation of the RNA, reverse transcription, PCR amplification) as well as by the bioinformatic algorithms used to detect novel isoforms. Using RNA spike-ins is a well-established strategy to solve -or at least alleviate- this problem (Hardwick et al., Nat Methods 2019: "spliced synthetic genes as internal controls in RNA sequencing experiments"). Thus, an additional run for WT and MUT, including spike-ins, should be performed to assess the robustness of the results. Secondly, the genome annotation used here is GENCODE v19, which is more than 7 years old, and comprises ~50K annotated transcripts, whereas the current annotation, GENCODE v33, comprises about 4 times more annotated transcripts. The authors should reanalyze their datasets with current annotations, specially considering that the detection of novel isoforms using long-read sequencing is one of the major claims stated in the abstract ("75% of isoforms do not overlap with GENCODE"). Thirdly, the replicability of the findings is not addressed throughout the paper. For example, is the differential usage of isoforms in UPP1 recapitulated in replicate 2? Why is the third (technical) replicate not used to validate the findings? How would the polysomal profiles of UPP1 and BUB3 look in independent biological replicates? Do the results reported in the manuscript hold in the third technical replicate if this one were independently analyzed (i.e. not used for the detection of candidate genes, only for validation)? If the third replicate would be analyzed independently, would the same sets of genes be identified as 'differential isoform usage' genes? How consistent are the results when different thresholds (e.g. using 5 reads or 3 reads) are used to define an isoform as "real"? Overall, the robustness of the results and the false discovery rate needs to be better addressed throughout the manuscript prior to considering this work for publication, in addition to using a more recent transcriptome annotation.

MAJOR COMMENTS

- 1- The authors claim in the abstract as one of their major findings the fact that "75% (49,366) of our long-read constructed multiexon isoforms do not overlap GENCODE or short-read assembled isoforms." However, the lack of spike-in controls in the sample does not allow to assess the False Discovery rate of isoform annotation and quantification. There are many different types of spike-ins that could be used to assess this. Given that the main point of the manuscript is the discovery of novel isoforms and quantification of differential isoform usage, in the case of this manuscript, is an essential control, as otherwise it is not possible to assess how many of these predictions could be inaccuracies of the bioinformatic algorithm or of the library preparation. This control is key to be able to ascertain whether the new predicted isoforms are in fact true positives or false positives.
- 2- The authors mention in page 6 that annotations from GENCODE version 19 were used, which corresponds to the genome assembly hg19. This annotation was released in 2013 (see: ftp://ftp.ncbi.nlm.nih.gov/genomes/genbank/vertebrate_mammalian/Homo_sapiens/all_assembly_versions/GCA_000001405.14_GRCh37.p13/GCA_000001405.14_GRCh37.p13_assembly_report.txt). Considering that a major point of the manuscript is the annotation of novel isoforms, and the advantage that long-read sequencing shows compared to short-read sequencing (for which the data was already available) the data of this work should be reanalyzed in the context of both a newer genome assembly and annotation (current release is not version 19, but version 33, <https://www.encodegenes.org/human/>). Importantly, the total number of annotated transcripts in the current GENCODE version 33 is 227,912 (<https://www.encodegenes.org/human/stats.html>), which is far from the ~40,000 transcripts annotated transcripts in GENCODE version 19 that is used in the work (number estimated from Figure 2B). The authors should assess their results using more recent annotations.
- 3- The authors nicely validate the differential isoform usage of UPP1 using agarose gels (Figure 4). They also identify additional 162 genes with differential isoform usage between the two cell lines. It would be important to validate some additional examples using agarose gels among the top 10 ranked candidates, in addition to the UPP1 example shown in Figure 4E.
4. As shown in Figure 1, the authors perform long-read cDNA sequencing with 3 replicates, 2 are biological, and a third one is technical. However, throughout the manuscript there is not much information with regards to whether the findings (novel isoforms, exon usage, isoform usage, etc) are independently supported by each of the replicates. Additional information regarding whether the findings are observed and are consistent in independent biological replicates, should be included throughout the manuscript.
5. The authors analyze polysome data to assess whether isoform dysregulation is associated with changes in translation. The authors conclude from this analysis (illustrated in Figure 6E and 6F) that "our data indicate a role for translational control through a splicing-dependent manner, and demonstrate distinct mechanisms of U2AF1 S34F for modulating translation control of genes through spliced isoform dysregulation". However, I am not convinced that this conclusion can be made based on the presented data. Firstly, it would be important to include data from independent biological replicates in Figure 6E and F to determine the robustness of the findings (the data from Palangat et al. includes 2 biological replicates). Secondly, it would be useful to report the translation changes that are observed in more than two genes, for example by taking at least the 10 top-ranked genes with differential isoform usage (e.g. those highlighted in Figure 4B), in the form of supplementary figures. The authors do state that "Our results showed that 66% (42/63) of genes with U2AF1 S34F-associated isoform changes also had a significant change in polysome profile (Methods)." However, illustrating these examples, as well as the robustness of the findings across replicates of the Palangat et al data should be included to support the conclusions.
6. How robust are the findings if the authors had used a different threshold to assess an isoform as "true", e.g. if instead of choosing 3 reads to identify an isoform, they would require 5 reads to identify an isoform as true? Would the top-ranked list of genes with differential isoform usage change?

MINOR COMMENTS

1. Page 6. The authors mention: "We found significant differences in expression for isoforms not contained in our set of high confident FLAIR isoforms (p-value <0.001; Figure 2B top panel)". How is this "high confident" set of FLAIR isoforms defined? How is the p-value computed and what is it exactly referred to? How many FLAIR isoforms were significant? Please include the number of "high confident" isoforms in the sentence.
- 2- It would be useful to provide the coordinates for LINC02879/USFM, which the authors mention is a putative lincRNA discovered by this work. Is this putative lincRNA may be perhaps already annotated in current GENCODE v33 annotations?
- 3- The authors state: "We manually examined long-reads aligned to USFM and found poly(A) tails, suggesting USFM supporting reads are not likely to be 3' end fragmented products.". However, it is well known that internal annealing in A-rich regions can also lead to initiation of reverse transcription from non-terminal polyA sites, which could explain the results observed. The authors should show the genome tracks of the region to show that these are not due to internal annealing of the polydT to A-rich regions. Again, having a set of spike-ins in the dataset would also allow to assess whether the authors see such types of phenomena in the spike-ins, which would contribute to clarify whether such phenomena are artefacts or true observations.
4. There is a typo in the name of the base-caller "Albacore", which should be "Albacore".
5. Methods. "Lung adenocarcinoma short-read data from The Cancer Genome Atlas (601 samples total) was downloaded from CGHub using gtdownload (Wilks et al., 2014). TCGA donors with multiple RNA-seq bams were filtered by date to only include the most recent RNA-seq bam (495 samples)." Why was this filtering done? Why not use 601 instead of 495?
6. Methods "Briefly, STAR junctions were kept if they contained at least 3 uniquely aligned in either both Mut1a and Mut1b samples or in both WT1 and WT2 samples." How many isoforms were predicted using the criteria of 3 uniquely aligned in each single replicate (e.g. only in Mut1 and only in Mut2, relative to how many were common in both? Could the technical replicate (sequenced in an independent flowcell) also used as validation (not prediction) of the isoforms, to assess the false discovery rate?
7. It would be appreciated if the authors would test a second algorithm to predict isoforms in addition to FLAIR, to assess the robustness of the results regardless of the bioinformatic algorithm used.

We thank the reviewers for their productive feedback. We have read and addressed their concerns as follows.

Reviewer #1 (Comments to the Authors (Required)):

GENERAL ASSESSMENT

Soulette et al describe and analyze a Nanopore-based, long-read sequencing approach to identifying isoforms that are differentially expressed or spliced in the absence or presence of the common U2AF1 S34F mutation. This represents the first published assessment of isoform-level consequences of U2AF1 mutations using long-read technologies, and as such, provides a novel and useful contribution to the field. The data are thoroughly described and analyzed and appear to be of high quality (for example, the high correlation ($r = 0.88$) between delta psi values computed with short-read versus long-read data is impressive given the much lower coverage available for long-read data).

We would like to highlight that, to our knowledge, this study is still the first isoform-level analysis of U2AF1 mutations using long-read technology.

MINOR COMMENTS

1. This manuscript's most important contribution is a database of isoforms in WT and U2AF1-mutant cells, which is presented in GTF format in Supplemental File 1. The utility of this database could be significantly improved with a few simple changes, such as separating out the internal IDs (presumably assigned by FLAIR) from Ensembl gene and transcript IDs and adding gene names from HUGO. The same suggestions apply to the Supplementary Tables. Ensembl gene and transcript ID matching, as well as CDS assignment, should also be fully described in the methods.

We thank the reviewer for their suggestion and have updated Supplemental File 1 and all relevant Supplemental Tables to have HUGO names as well as separate Ensembl names. We have also added the following to the Methods, explaining how genes and transcripts were matched to Ensembl as part of the FLAIR pipeline.

“Gencode v19 was provided to FLAIR, which matched detected isoforms to the annotation based on the intron chain.”

2. The comparison to Gencode makes sense given the isoform-level information provided by long-read sequencing and FLAIR. However, the analysis of annotated versus unannotated junctions in Fig. 1 would be strengthened by a comparison to a more comprehensive junction database, such as MISO's isoform annotation.

We thank the reviewer for their feedback, which we believe pertains to Figure 2a. However, MISO's database specifically provides event-level annotations, such as exon skipping, derived

from GENCODE. It also doesn't provide any full-length isoform annotations. Therefore, a further comparison of our data to MISO would not strengthen our analysis. However, we now include a comparison of GENCODE (v19 and v33), RefSeq and UCSC Genes annotations to further compare our isoforms with additional transcript annotations.

3. The UCSC Genome Browser view in Fig. 4C is too small to be understandable.

We have altered Figure 4C to contain a more readable genome browser view. With this figure, we seek to demonstrate the diversity of possible isoforms we can detect for a single gene.

4. Because the experiments were conducted in unperturbed (not NMD-inhibited) cells, an alternative possibility is that NMD efficiency differs between the cell types. This seems unlikely, but nevertheless should be mentioned as a possible caveat when interpreting the data for Fig. 6.

In our Discussion section, we mention the possibility of differences in NMD efficiency and also cite a more recent study Cheruiyot et al. 2021 that suggests that U2AF1 mutation globally alters NMD.

Reviewer #2 (Comments to the Authors (Required)):

The manuscript by Soulette et al, entitled "Nanopore sequencing reveals U2AF1 S34F-associated full-length isoforms" reports long-read RNAseq results from U2AF1 mutant HBEC3kt cells. The authors identified isoforms not detected by short-read RNAseq and highlight how alternative splicing of two genes may affect translational activity.

Major:

1. Please add the level of expression of LINC02879 from long-read data for MT and WT samples.

We added the following sentence to the portion of the paper where we introduce USFM:

We investigated a putative lncRNA we call USFM (upregulated in splicing factor mutant; LINC02879; chr 18:26,735,945-26,754,735), which was one of the most highly expressed multi-exon isoforms in mutant samples with 202 reads per million (RPM) (17 RPM in wild type; Figure 2D bottom panel).

2. Please include the overlap in junctions between long-read and short-read data for Fig. 3A (225 vs. 142 junctions). Could the authors also provide the overlap between short-read TCGA and short-read HBEC3kt.

The Fei paper from which the short-read HBEC3kt was obtained has an overlap analysis of that data with the TCGA. We have generated an additional supplemental table (ST7) which contains the overlap of cassette exon events identified in the Fei short-read data, that were found to be significantly altered from our long-read data in this study. The change and percent spliced in values from our study and the Fei study have been included. We have also included supplemental table 8 which contains an overlap of juncBASE cassette exon events between TCGA LUAD data and HBEC3kt short-read data. During the revision, we double-checked the analysis and now it correctly shows the number of events with splicing changes with a delta value > 10% PSI.

3. Can the authors provide a clear explanation in the text for the difference between isoform usage (ratio of isoform expression within a gene) versus the isoform expression (absolute level of expression). This will be important to understand Fig. 5 and the UPP1 results.

We added the following sentence to the section *U2AF1 S34F induces strong isoform switching in UPP1 and BUB3*

“We tested for changes in both i) the ratio of isoform expression within a gene (isoform usage), and ii) the absolute level of expression (isoform expression).”

4. Fig. 5A and 5C. It appears that the majority of the DIE are associated with DGE in Fig 5B and 5C, suggesting that most of the isoform expression changes are due to gene expression changes and may not be as informative as the isoform usage data. Can the authors comment on this.

We thank the reviewer for highlighting this point, which we have now included in manuscript, for example in describing the results shown in Figure 5B:

“Most of these isoform changes were associated with total gene expression changes (Figure 5B), suggesting these are transcriptionally regulated.”

5. UPP1 had increased productive isoforms in mutant cells, but significant downregulation of expression. This seems counterintuitive, can the authors suggest a reason? Can the authors provide another gene example of a productive isoform but downregulation of expression? Is this an isolated gene, or is it common in U2AF1 mutant cells?

We thank the reviewer for their suggestion. We performed the suggested analysis and found that this is uncommon. Out of the 9 genes with substantial changes in PTC isoform usage, UPP1 was the only gene with significant changes in gene expression. In our discussion, we describe the possibility that U2AF1 mutations affect NMD efficiency and there is previous literature to support this; however, further studies are necessary to explain this mechanism.

6. Are there other gene examples of APA shifts that lead to expression changes in polysome fractions? Is BUB3 an isolated example, or is this finding common in U2AF1 mutant cells?

Thank you for suggesting this additional analysis. Of the 10 genes with significant changes in APA, 6 were associated with strong polysome profile shifts toward higher polysome fractions, 3 showed a shift toward monosome fractions, and 1 remained the same across WT and MT conditions. We have now included this description in the main manuscript.

Minor:

1. Please include in the main text the summary statistics for the number of reads and read length per sample for the Nanopore data and the short-read data from HBEC3kt cells.

We have added the Nanopore summary statistics in Supplementary Table 1 and included the short-read statistics in the text *“For comparison to the long-read data, the short-read data were approximately 100 million paired-end 101bp reads per sample.”*

2. On page 6, the authors hypothesized that the increased number of annotated isoforms from short-read assembly could be due to higher sequencing depths. It is unclear if the results in Fig 2B support their hypothesis. Please provide the answer in the text.

We have updated the text and associated figure to improve clarity in this section. The following text has been added:

“Indeed, we found that the expression of annotated isoforms found exclusively by short-read assembly had significantly lower expression than any isoforms identified by long-reads ...”

We have also included text in the discussion to address isoform discovery limitation using long-read (supplemental figure 6).

In line with these shortcomings, a saturation analysis of full-length isoform construction reveals isoform discovery limitations, possible due to relatively shallow sequencing depth (**Supplemental Figure 6**).

3. How reproducible are the long-read results based on a technical replicate - were the results from MT2 B1 and MT2 B2 samples more similar to each other than MT1 B1?

We thank the reviewer for this question. We have added an additional supplemental figure to address the isoform overlap between our replicates (Supplemental Figure 1D). We also took this opportunity to revise our language when describing the sequencing scheme. To clarify,

we did not include any technical replicates in this analysis, so we cannot speak to the overlap between such replicates. All of our replicates were independent growth replicates. We have adjusted the text to better reflect this:

“We obtained 3 biological replicates for each WT and MT condition by extracting whole-cell RNA from each cell isolate, one growth replicate of WT1 and MT1 and two independent growth replicates from different time points for WT2 and MT2.”

4. Why does LINC02879 need to be renamed USFM by the authors? Why not just call it LINC02879 to avoid confusion.

We believe that USFM is a more useful functional name in the context of our study and will help the readability in sections where we refer to it. Since we consistently refer to it as USFM throughout the paper, we don't believe this name will introduce additional reader confusion. Additionally, it is quite common for lncRNAs to have alias based on any known function or regulation (e.g. MALAT1 - Metastasis Associated Lung Adenocarcinoma Transcript 1 - LINC00047)

5. On the top of page 15, "198 isoforms with altered usage and expression", I think should read "usage or expression" not "and".

We have changed this sentence to reflect your suggestions.

6. Fig. 4B. The two green colors are very similar in panel D. Is there an alternative color or style?

We understand your concern and have updated Figure 4 with one of the greens changed to a darker shade of green.

7. Fig 5C. Please highlight the UPP1 gene in this panel

We have updated Figure 5C to highlight UPP1. It is indicated with a red arrow and gold bars.

8. In the discussion, please add additional information on why UPP1 is important - what is its normal function, etc.

We have added the following sentence to the discussion:

“UPP1 encodes a uridine phosphorylase, which helps maintain homeostatic levels of uridine for RNA synthesis and other processes.”

9. Can the authors please provide more information on the FLAIRcorrect method here?

The FLAIR correct portion of the methods section has been updated to include the following sentence, which elaborates on how this module works:

“The short-read junctions, along with GENCODE annotated junctions, were used to correct misaligned splice sites in the nanopore data to the nearest site within 10bp.”

In addition, since the original submission, the primary manuscript describing FLAIR was published in Nature Communications in 2020 (Tang et al. Nat Comm 2020).

10. Supp file 3. Please add gene symbols.

We have added the HUGO gene symbols to this file.

11. Supp Table 2. Please add the average or median read counts for each group. Alternatively, provide the read counts per sample if there are not too many samples.

We have added the average read counts for each group, as well as the number of samples in each group in the first row of the table.

12. Supp Table 3. Please add the mutation status of each sample.

We have updated the table so that each sample's name includes its mutation status and other relevant information.

13. Supp Table 4. Please add the gene symbol and read counts.

We have added an additional column containing the HUGO gene symbol and counts for each WT and MUT sample used for the analysis.

14. Supp Table 5. Please add values for each sample.

We have uploaded the raw counts table (SUPPLEMENTAL FILE 3) and added additional columns containing raw counts for each sample.

15. Supp Table 6. Please add the gene symbols and values for each sample.

We have added the HUGO gene symbols to both sheets and the values for each sample can be referred to in the sheet we added to table 5.

Reviewer #3 (Comments to the Authors (Required)):

SUMMARY

In this work, Soulette and colleagues explore the isoform alterations that result from the mutation U2AF1 S34F, which is a recurrent mutation in lung adenocarcinoma. To this end, the authors conduct high-throughput long-read cDNA sequencing using nanopore technologies, comparing HBEC3kt cells (WT) to those that contain the mutation (MUT). Using this strategy, the authors report 75% of novel isoforms that do not overlap with GENCODE annotations. They then identify genes with significant isoform usage differences, as well as pursue additional analyses to address the consequences of isoform usage, such as changes in the translational profiles. While the rationale of the work is correct and the question is relevant, I find that there are essential controls that are missing in this work, and without them it is impossible to evaluate whether the findings are correct. Firstly, and perhaps this is the most important point of all, the authors do not include RNA spike-ins (e.g. sequins, SIRVs) in their samples that could serve to assess the false discovery rate of isoforms or to assess the robustness of their bioinformatic pipeline in the analyses of their data. While nanopore technologies are capable of detecting full-length mRNA isoforms, there are plenty of artefacts that can affect the detection of novel isoforms, which can be caused by the library preparation (fragmentation of the RNA, reverse transcription, PCR amplification) as well as by the bioinformatic algorithms used to detect novel isoforms. Using RNA spike-ins is a well-established strategy to solve -or at least alleviate- this problem (Hardwick et al., Nat Methods 2019: "spliced synthetic genes as internal controls in RNA sequencing experiments"). Thus, an additional run for WT and MUT, including spike-ins, should be performed to assess the robustness of the results. Secondly, the genome annotation used here is GENCODE v19, which is more than 7 years old, and comprises ~50K annotated transcripts, whereas the current annotation, GENCODE v33, comprises about 4 times more annotated transcripts. The authors should reanalyze their datasets with current annotations, specially considering that the detection of novel isoforms using long-read sequencing is one of the major claims stated in the abstract ("75% of isoforms do not overlap with GENCODE"). Thirdly, the replicability of the findings is not addressed throughout the paper. For example, is the differential usage of isoforms in UPP1 recapitulated in replicate 2? Why is the third (technical) replicate not used to validate the findings? How would the polysomal profiles of UPP1 and BUB3 look in independent biological replicates? Do the results reported in the manuscript hold in the third technical replicate if this one were independently analyzed (i.e. not used for the detection of candidate genes, only for validation)? If the third replicate would be analyzed independently, would the same sets of genes be identified as 'differential isoform usage' genes? How consistent are the results when different thresholds (e.g. using 5 reads or 3 reads) are used to define an isoform as "real"?

Overall, the robustness of the results and the false discovery rate needs to be better addressed throughout the manuscript prior to considering this work for publication, in addition to using a more recent transcriptome annotation.

MAJOR COMMENTS

1- The authors claim in the abstract as one of their major findings the fact that "75% (49,366) of our long-read constructed multiexon isoforms do not overlap GENCODE or short-read

assembled isoforms " However, the lack of spike-in controls in the sample does not allow to assess the False Discovery rate of isoform annotation and quantification. There are many different types of spike-ins that could be used to assess this. Given that the main point of the manuscript is the discovery of novel isoforms and quantification of differential isoform usage, in the case of this manuscript, is an essential control, as otherwise it is not possible to assess how many of these predictions could be inaccuracies of the bioinformatic algorithm or of the library preparation. This control is key to be able to ascertain whether the new predicted isoforms are in fact true positives or false positives.

We believe that the FDR for novel isoforms produced by FLAIR was addressed in our publication that was published subsequent to our first submission (Tang et al. Nature Communications 2020). This issue is specifically addressed in Tang et al. Figure 2B which we include below. In the publication we compared our tool to two other competing methods and found that FLAIR performed better or equal to these others.

We evaluated FLAIR against both RNA spike-in SIRV data, simulated data, and real data (primary B cell sequencing) and as you can see, depending on the dataset, we show a precision that FLAIR has sensitivity between ~28-80% and precision between ~90-98%, but always comparable or better than other competing tools.

As also suggested by reviewers, we also ran our data through another method StringTie2 and show that our general results are robust to the transcriptome analysis method performed (see response to Minor Comment #7 below)

Additionally, we have adjusted the language in our abstract to better reflect that the majority isoforms discovered in this study simply do not match annotations and short-read assembly.

2- The authors mention in page 6 that annotations from GENCODE version 19 were used, which corresponds to the genome assembly hg19. This annotation was released in 2013 (see: ftp://ftp.ncbi.nlm.nih.gov/genomes/genbank/vertebrate_mammalian/Homo_sapiens/all_assembly_versions/GCA_000001405.14_GRCh37.p13/GCA_000001405.14_GRCh37.p13_assembly_report.txt). Considering that a major point of the manuscript is the annotation of novel isoforms, and the advantage that long-read sequencing shows compared to short-read sequencing (for which the data was already available) the data of this work should be reanalyzed in the context of both a newer genome assembly and annotation (current release is not version 19, but version 33, <https://www.encodegenes.org/human/>). Importantly, the total number of annotated transcripts in the current GENCODE version 33 is 227,912 (<https://www.encodegenes.org/human/stats.html>), which is far from the ~40,000 transcripts annotated transcripts in GENCODE version 19 that is used in the work (number estimated from Figure 2B). The authors should assess their results using more recent annotations.

We appreciate this point that the reviewer addresses and would like to recognize the lack of clarity in Figure 2b that raises this question. We feel that we have addressed this concern by remaking Figure 2 to include the overlap between FLAIR and a merged annotation set made up of gencode 19, gencode 33 mapped to GRCh37, UCSC genes, and refseq. This both ensures that the overlap includes more recently annotated isoforms and reduces annotation bias in our set of unique FLAIR isoforms. We initially did the analysis because the short read data we used from Fei et al. and ~450 TCGA lung adenocarcinoma data was aligned to hg19, so doing our analysis primarily using hg19 allowed better comparison to these matched data sets.

3- The authors nicely validate the differential isoform usage of UPP1 using agarose gels (Figure 4). They also identify additional 162 genes with differential isoform usage between the two cell lines. It would be important to validate some additional examples using agarose gels among the top 10 ranked candidates, in addition to the UPP1 example shown in Figure 4E.

We agree with the reviewer that RT-PCR validation of more targets will strengthen our results from FLAIR. We have now included data for additional 7 targets as supplemental Figure 3.

4. As shown in Figure 1, the authors perform long-read cDNA sequencing with 3 replicates, 2 are biological, and a third one is technical. However, throughout the manuscript there is not much information with regards to whether the findings (novel isoforms, exon usage, isoform usage, etc) are independently supported by each of the replicates. Additional information regarding whether the findings are observed and are consistent in independent biological replicates, should be included throughout the manuscript.

We thank the reviewer for raising concerns regarding replicate reproducibility. As previously mentioned, the samples from each condition were treated as independent biological replicates, given that their genetic backgrounds are the same within each condition. To address consistency between replicates, our statistical approach considers the expression variability within replicates and compares it to the variability between conditions. Therefore, the p-values

we report for expression and usage should already reflect the consistency of measured values across replicates. The cDNA from the “third replicate” is from an independent growth replicate. Our definition of a technical replicate is a pair of sequencing data generated from the same cDNA, which is not what we’ve done. The following sentence was added to the paper to clarify this point:

“We obtained 3 biological replicates for each WT and MT condition by extracting whole-cell RNA from each cell isolate, one growth replicate of WT1 and MT1 and two independent growth replicates from different time points for WT2 and MT2.”

To address the concern about consistency between replicates, we have added a supplemental figure 1D with a venn diagram showing the overlap of detected isoforms between replicates for each condition. These venn diagrams show strong correlation of detected isoforms between replicates. As to the reviewer’s earlier question about the consistency in differential isoform usage between replicates, we have found that while the two samples from batch 1 have slightly more DIU than the sample from batch 2, all samples have significant DIU.

5. The authors analyze polysome data to assess whether isoform dysregulation is associated with changes in translation. The authors conclude from this analysis (illustrated in Figure 6E and 6F) that "our data indicate a role for translational control through a splicing-dependent manner, and demonstrate distinct mechanisms of U2AF1 S34F for modulating translation control of genes through spliced isoform dysregulation". However, I am not convinced that this conclusion can be made based on the presented data. Firstly, it would be important to include data from independent biological replicates in Figure 6E and F to determine the robustness of the findings (the data from Palangat et al. includes 2 biological replicates). Secondly, it would be useful to report the translation changes that are observed in more than two genes, for example by taking at least the 10 top-ranked genes with differential isoform usage (e.g. those highlighted in Figure 4B), in the form of supplementary figures. The authors do state that "Our results showed that 66% (42/63) of genes with U2AF1 S34F-associated isoform changes also had a significant change in polysome profile (Methods)." However, illustrating these examples, as well as the robustness of the findings across replicates of the Palangat et al data should be included to support the conclusions.

We understand the reviewer’s concern about the replicability of this result. To address this we have included a number of additional polysome profiles in Supplemental Figure 5 from Palangat et. al. We have included genes with significant isoform changes (gray) and genes with significant changes in 3’ polyA processing (gold). We have also adjusted the language to more accurately describe the associations we find to be consistent with reports from Palangat et. al. The replicate count information however was not included in the final Palanagat supplemental data, but we have included the authors negative control data in the additional polysome profile plots included in Supplemental Figure 5.

6. How robust are the findings if the authors had used a different threshold to assess an isoform as "true", e.g. if instead of choosing 3 reads to identify an isoform, they would require 5 reads to identify an isoform as true? Would the top-ranked list of genes with differential isoform usage change?

We thank the reviewer for raising this question regarding long-read isoform construction parameters, and their impact on downstream analyses. Most confident changes in expression and or usage come from isoforms with many more than 5 reads, as the number of reads is something that FLAIR takes into account when determining the confidence level of an isoform. This question was also addressed in our Tang et al. Nature Communications 2020 paper in which the FLAIR isoform detection method is described in more detail. In Figure 2B from that paper (below), we showed how different read thresholds affect the sensitivity and precision of FLAIR. As also suggested below, we also used an alternative pipeline for isoform detection from long-reads, StringTie2, and our results are robust to the isoform detection approach (more detailed below).

MINOR COMMENTS

1. Page 6. The authors mention: "We found significant differences in expression for isoforms not contained in our set of high confident FLAIR isoforms (p-value <0.001; Figure 2B top panel)". How is this "high confident" set of FLAIR isoforms defined? How is the p-value computed and what is it exactly referred to? How many FLAIR isoforms were significant? Please include the number of "high confident" isoforms in the sentence.

We thank the reviewer for their interest in the methods of the FLAIR pipeline. We acknowledge that the language about high confidence isoforms from FLAIR was confusing. All isoforms that FLAIR returns are high confidence, so we have removed that language from the paper. FLAIR does not return a p-value for its isoforms. The p-value the reviewer referenced was in relation to the comparison between isoforms that intersect between the FLAIR and String-Tie datasets and those unique to a dataset. We have updated the sentence to be more clear:

"We found a significant difference in the average expression of isoforms exclusive to StringTie relative to FLAIR detected isoforms (p-value <0.01; Figure 2C)."

We have also updated Figure 2 to more clearly reflect the data.

2- It would be useful to provide the coordinates for LINC02879/USFM, which the authors mention is a putative lincRNA discovered by this work. Is this putative lincRNA may be perhaps already annotated in current GENCODE v33 annotations?

We have added the coordinates in the paper when introducing USFM and have checked a more current GENCODE annotation (GENCODE v41) at this locus and have observed no other annotations.

3- The authors state: "We manually examined long-reads aligned to USFM and found poly(A) tails, suggesting USFM supporting reads are not likely to be 3' end fragmented products.". However, it is well known that internal annealing in A-rich regions can also lead to initiation of reverse transcription from non-terminal polyA sites, which could explain the results observed. The authors should show the genome tracks of the region to show that these are not due to internal annealing of the polydT to A-rich regions. Again, having a set of spike-ins in the dataset would also allow to assess whether the authors see such types of phenomena in the spike-ins, which would contribute to clarify whether such phenomena are artefacts or true observations.

We thank the reviewer for their suggestions for validating USFM. There is a filtering step in FLAIR that asks if there is a poly-A or poly-T that isn't soft-clipped, indicating internal priming, then tosses out those reads. We have also added a supplementary figure with the genome browser track for USFM, which shows that this region of the genome is not highly repetitive. While the second exon overlaps a repeat region, the region as a whole is not repetitive. However, that overlap of exon 2 with a repetitive region may explain why this lincRNA has not been detected through short read methods, as graphical transcriptome assembly methods struggle to assemble across low complexity regions.

4. There is a typo in the name of the base-caller "Albabacore", which should be "Albacore".

Thank you for catching this typo. We have fixed it.

5. Methods. "Lung adenocarcinoma short-read data from The Cancer Genome Atlas (601 samples total) was downloaded from CGHub using gtdownload (Wilks et al., 2014). TCGA donors with multiple RNA-seq bam files were filtered by date to only include the most recent RNA-seq bam (495 samples). " Why was this filtering done? Why not use 601 instead of 495? Our analysis did not consider the longitudinal effect of *U2AF1* S34F mutation, and therefore timepoint information was not considered in any of our statistical approaches. The selection of the most recent timepoint was used as an unbiased approach to remove potential confounding factors that may affect our statistical analysis.

6. Methods "Briefly, STAR junctions were kept if they contained at least 3 uniquely aligned in either both Mut1a and Mut1b samples or in both WT1 and WT2 samples." How many isoforms

were predicted using the criteria of 3 uniquely aligned in each single replicate (e.g. only in Mut1 and only in Mut2, relative to how many were common in both? Could the technical replicate (sequenced in an independent flowcell) also used as validation (not prediction) of the isoforms, to assess the false discovery rate?

To address your valid concerns about the variability between samples, we have updated Supplementary Figure 1 with a venn diagram showing the overlap in detected isoforms between samples. As we have already mentioned, all three replicates are considered biological, not technical. As for FDR, we believe that this has already been addressed by our answer to Major Comment #6.

7. It would be appreciated if the authors would test a second algorithm to predict isoforms in addition to FLAIR, to assess the robustness of the results regardless of the bioinformatic algorithm used.

We recognize the importance in demonstrating robustness of bioinformatic methods, but we believe that a side-by-side comparison of other isoform detection methods from long-reads is beyond the scope of our manuscript. In-fact we are significant contributors to an international consortium called the Long-Read RNA-Seq Genome Annotation Assessment Project (LRGASP) to perform a more systematic comparison of isoform prediction tools which has recently been accepted in-principle at Nature Methods.

Nevertheless, we analyzed our long-read data using an alternative method for identifying long-read isoforms - StringTie2 (version 2.15). We also found predominant cassette exon changes associated with U2AF1 mutation which demonstrates the robustness of our methods applied in this manuscript. This result is now included in Supplemental Figure 1F.

June 12, 2023

RE: Life Science Alliance Manuscript #LSA-2020-00641-TR

Dr. Angela N. Brooks
University of California, Santa Cruz
1156 High Street M/S: SOE2
Santa Cruz, CA 95064

Dear Dr. Brooks,

Thank you for submitting your revised manuscript entitled "Full-length transcript alterations in human bronchial epithelial cells with U2AF1 S34F mutations". We would be happy to publish your paper in Life Science Alliance pending final revisions necessary to meet our formatting guidelines.

- please address Reviewer 2's remaining comments
- please add Keywords for your manuscript to our system
- please add the Twitter handle of your host institute/organization as well as your own or/and one of the authors in our system
- please add your main, supplementary figure, and table legends to the main manuscript text after the references section
- all figure legends should only appear in the main manuscript file
- please add callouts for Tables S2 and S3
- please be sure to refer to all supplementary material in the manuscript text
- please add a callout for Figure S3 accordingly

A. FINAL FILES:

B. MANUSCRIPT ORGANIZATION AND FORMATTING:

Sincerely,

Reviewer #2 (Comments to the Authors (Required)):

I thank the authors for their responses. There are several points for clarity that need to be addressed.

1. The supplemental figures did not have figure numbers on them. Please add supplemental figure numbers.
2. Supplemental figure 3 is missing the 'Z' label. Also, please label the 3 panels after panel Z with another label instead of 2A, 2B, 2C. This is confusing.
3. Please label which isoforms in UPP1 are PTC-containing isoforms. The authors suggest that U2AF1-S34F reduces NMD isoforms in UPP1, but this is not clear from the figure.
4. The conclusions drawn from the monosome and polysome data is overreaching and not clear in its current form. As an example, in Fig 6E, the authors state there is a marked shift of UPP1 expression in mutant samples from the monosome toward higher polysome (\geq fraction 7) in mutant samples. However, fractions 7 and 8 are slightly higher (but $<10\%$ shift) and fractions 9 and 10 are lower in the mutant. I suggest that the polysome fraction data could all be moved supplemental or removed from the manuscript.
5. Supplemental table 4. The authors added readcounts, but it appears there is a formatting issue when the files open they do not have read data in the sample columns.

Response to Reviewer

We would like to thank the reviewer for the additional helpful comments and have addressed all remaining issues.

Reviewer #2 (Comments to the Authors (Required)):

I thank the authors for their responses. There are several points for clarity that need to be addressed.

1. The supplemental figures did not have figure numbers on them. Please add supplemental figure numbers.

Since we are submitting final figures, we have not included figure numbers on the Supplemental Figures as they will be submitted individually. We have now ensured that all Supplemental figure legends are in the main manuscript and are referenced in the text.

2. Supplemental figure 3 is missing the 'Z' label. Also, please label the 3 panels after panel Z with another label instead of 2A, 2B, 2C. This is confusing.

We have now changed those labels to AA, AB, and AC, accordingly.

3. Please label which isoforms in UPP1 are PTC-containing isoforms. The authors suggest that U2AF1-S34F reduces NMD isoforms in UPP1, but this is not clear from the figure.

For increased clarity, we have now colored the isoform labels consistently to indicate PTC vs PRO (productive) to make more clear which isoforms are PTC-containing.

4. The conclusions drawn from the monosome and polysome data is overreaching and not clear in its current form. As an example, in Fig 6E, the authors state there is a marked shift of UPP1 expression in mutant samples from the monosome toward higher polysome (\geq fraction 7) in mutant samples. However, fractions 7 and 8 are slightly higher (but $<10\%$ shift) and fractions 9 and 10 are lower in the mutant. I suggest that the polysome fraction data could all be moved supplemental or removed from the manuscript.

We have now moved these figure panels to the Supplemental Figures. These are now figures S5B-D.

5. Supplemental table 4. The authors added readcounts, but it appears there is a formatting issue when the files open they do not have read data in the sample columns.

Thank you for finding this error. This was due to converting tables to Excel. We have fixed the formatting issues and have double checked Excel conversion to all other tables.

July 3, 2023

RE: Life Science Alliance Manuscript #LSA-2020-00641-TRR

Dr. Angela N. Brooks
University of California, Santa Cruz
1156 High Street M/S: SOE2
Santa Cruz, CA 95064

Dear Dr. Brooks,

Thank you for submitting your Research Article entitled "Full-length transcript alterations in human bronchial epithelial cells with U2AF1 S34F mutations". It is a pleasure to let you know that your manuscript is now accepted for publication in Life Science Alliance. Congratulations on this interesting work.

DISTRIBUTION OF MATERIALS:

Again, congratulations on a very nice paper. I hope you found the review process to be constructive and are pleased with how the manuscript was handled editorially. We look forward to future exciting submissions from your lab.

Sincerely,
